# Visualizing the internal structure of the charge-density-wave state in CeSbTe

Xinglu Que[1], Qingyu He[1], Lihui Zhou[1], Shiming Lei[2,5], Leslie Schoop [2],
Dennis Huang [1] ✉ & Hidenori Takagi [1,3,4]

The collective reorganization of electrons into a charge density wave has long served as a textbook example of an ordered phase in condensed matter physics. Two-dimensional square lattices with $p$ electrons are well-suited to the realization of charge density waves, due to the anisotropy of the $p$ orbitals and the resulting one dimensionality of the electronic structure. In spite of a long history of study of charge density waves in square-lattice systems, few reports have recognized the significance of a hidden orbital degree of freedom. The degeneracy of $p_x$ and $p_y$ electrons may give rise to orbital patterns in real space that endow the charge density wave with additional broken symmetries or unusual order parameters. Here, we use scanning tunneling microscopy to visualize the internal structure of the charge-density-wave state of CeSbTe, which contains Sb square lattices with $5p$ electrons. We image atomic-sized, anisotropic lobes of charge density with periodically modulating anisotropy, which we interpret in terms of a superposition of $p_x$ and $p_y$ bond density waves. Our results support the fact that delocalized $p$ orbitals can reorganize into emergent electronic states of matter.

The $p$-electron square lattice represents a prototypical model for charge density wave (CDW) in two dimensions (2D)[1–4]. As shown in Fig. 1a, we consider degenerate, half-filled $p_x$ and $p_y$ orbitals, where the $(x', y')$ axes are defined along the nearest-neighbor bonds with length $a'$. Due to the anisotropy of the $p$ orbitals, σ hopping is much larger than π hopping, leading $p_x$ electrons to propagate predominantly along the $x'$ direction and $p_y$ electrons to propagate predominantly along the $y'$ direction. This anisotropy is reflected in the Fermi surface, which comprises quasi-1D $p_x$ bands centered around crystal momentum $k_{x'} = \pm\pi/(2a')$ and quasi-1D $p_y$ bands centered around $k_{y'} = \pm\pi/(2a')$.

In the limit of only σ hopping, $t_\sigma$, the $p_x$ and $p_y$ bands are fully 1D and we can consider a few possible CDW states with different wave vectors (Fig. 1b). A wave vector $\mathbf{q} = (\pi/a', 0)$ nests the $p_x$ bands and corresponds in real space to a bond density wave of $p_x$ electrons with periodicity $2a'$ along the $x'$ axis (Fig. 1c). This bond density wave can be viewed as parallel chains of dimerized $p_x$ orbitals with σ overlap and is analogous to the Peierls distortion of a 1D chain of $s$ orbitals.

Equivalently, a wave vector $\mathbf{q} = (0, \pi/a')$ nests the $p_y$ bands and gives rise to a bond density wave of $p_y$ electrons with periodicity $2a'$ along the $y'$ axis (Fig. 1d). Another wave vector $\mathbf{q} = (\pi/a', \pi/a')$ simultaneously nests both the $p_x$ and $p_y$ bands. In real space, the corresponding CDW state is one where the neighboring chains of dimerized $p_x$ and $p_y$ orbitals are phase-shifted by $a'$ relative to one another, resulting in a $\sqrt{2}a' \times \sqrt{2}a'$ supercell rotated by 45° relative to the primitive unit cell (Fig. 1e). The phase shift incurs no additional energy cost, as neighboring chains of dimers are independent in the 1D limit we consider, and instead facilitates additional energy gain by allowing the $p_x$ and $p_y$ bands to be simultaneously gapped with a single $\mathbf{q}$.

An overlooked feature of this textbook model is the underlying orbital degree of freedom. In the case of the $\mathbf{q} = (\pi/a', \pi/a')$ CDW state, we essentially have independent $p_x$ and $p_y$ bond density waves, and the degree of freedom concerns how these two should be superimposed in real space. One such superposition results in extended zigzag bonds of $p$ orbitals that run along the [11] direction (Fig. 1f).

[1]Max Planck Institute for Solid State Research, Stuttgart, Germany. [2]Department of Chemistry, Princeton University, Princeton, NJ, USA. [3]Institute for Functional Matter and Quantum Technologies, University of Stuttgart, Stuttgart, Germany. [4]Department of Physics, University of Tokyo, Tokyo, Japan. [5]Present address: Department of Physics, Hong Kong University of Science and Technology, Hong Kong, China. ✉e-mail: D.Huang@fkf.mpg.de

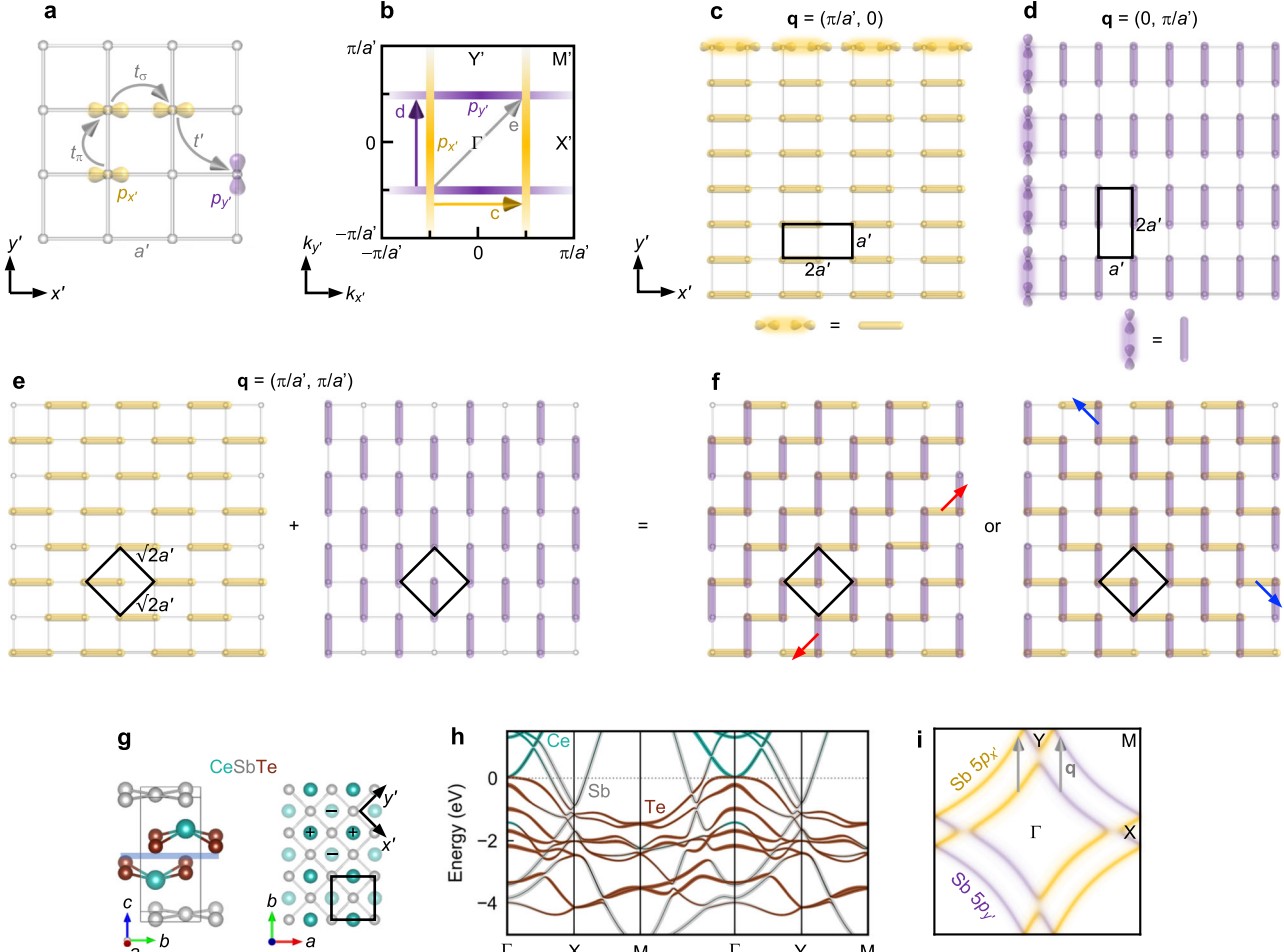

**Fig. 1 | $p$-electron phases on a square lattice and CeSbTe. a** Tight-binding model of $p_{x'}$ and $p_{y'}$ electrons on a square lattice with bond length $a'$ and hopping parameters $t_\sigma$, $t_\pi$, and $t'$. The axes ($x'$, $y'$) are defined along the bond directions of the square lattice. **b** Schematic Fermi surface in the limit of $t_\sigma \gg t_\pi \approx t' \approx 0$. Yellow (purple) denotes the contribution from $p_{x'}$ ($p_{y'}$) orbitals. **c** Bond density wave of $p_{x'}$ electrons with wave vector $\mathbf{q} = (\pi/a', 0)$, shown as a yellow arrow in (**b**). The supercell is enclosed in a black box. **d** Bond density wave of $p_{y'}$ electrons with wave vector $\mathbf{q} = (0, \pi/a')$, shown as a purple arrow in (**b**). **e** The wave vector $\mathbf{q} = (\pi/a', \pi/a')$ simultaneously nests both $p_{x'}$ and $p_{y'}$ Fermi sheets (gray arrow in **b**) and results in both $p_{x'}$ (left subpanel) and $p_{y'}$ (right subpanel) bond density waves. **f** The superposition of $p_{x'}$ and $p_{y'}$ bond density waves results in zigzag bond patterns that

extend either along the [11] (left subpanel) or [$\bar{1}$1] (right subpanel) directions, depending on the relative phase of the two density waves. **g** Crystal structure of CeSbTe. The ($a$, $b$) axes are rotated 45° from the ($x'$, $y'$) axes. The blue plane indicates where cleavage takes place between two CeTe layers. The Ce atoms labeled (+) and (−) lie above and below the Sb square lattice, respectively, which enlarges the unit cell to contain two Sb atoms. **h** Calculated band structure of $\mathrm{Ce(Sb_{1-x}Te_x)Te}$ decomposed into corresponding contributions from Ce, Sb, and Te. The virtual crystal approximation was applied to simulate a Te substitutional level of $x = 0.37$ (see Supplementary Note 2 for estimation of doping level). **i** Tight-binding approximation of the Fermi surface with $p$-orbital characters and CDW wave vectors shown.

Another such superposition with a different phase relationship between the two density waves results in extended zigzag bonds that run along the [$\bar{1}$1] direction. Here, the two possible superpositions are trivially related by a 90° rotation. If we have $p_{x'}$ and $p_{y'}$ bond density waves with longer wavelengths and more complicated patterns, superimposing the two density waves with different phase relationships may result in actually distinct patterns. In reality, the dimerized chains of $p_{x'}$ and $p_{y'}$ orbitals are not fully independent, but interact via interchain hopping ($t_\pi$) and interorbital hopping ($t'$; see Fig. 1a), which then select the real-space structure and relative phase of $p_{x'}$ and $p_{y'}$ bonds within the CDW state.

$R$Te$_3$, $R$Te$_2$, and $R$SbTe ($R$ = rare earth element) are related material classes that host 2D square lattices of pnictogen or chalcogen atoms with $p$ electrons sandwiched by (generally) magnetic buffer layers. While the study of CDW in these materials has an extensive history[1,5–12], only recently have any experimental signatures emerged indicating a nontrivial orbital texture in the CDW state. Raman spectroscopy measurements probing the collective amplitude mode of the CDW in GdTe$_3$ and LaTe$_3$ detected an unexpected axial symmetry[13],

which points to an unconventional order parameter for the CDW, possibly with finite orbital angular momentum[14–16]. The microscopic orbital pattern that is inferred to be responsible for this state, however, has not yet been directly imaged.

We focus on CeSbTe[17], which has drawn renewed interest as a material platform that offers Dirac-like bands tunable by magnetism and CDW[18,19]. The compound is composed of Sb square lattices separated by CeTe layers (Fig. 1g). Due to the staggered arrangement of Ce atoms above and below the Sb plane, the Sb square lattice acquires nonsymmorphic symmetry and has doubled in-plane periodicity, with two Sb atoms per unit cell. The folding of the Sb $5p$ bands creates linearly dispersing bands with an approximate Dirac line node near the Fermi energy, whose diamond shape reflects the quasi-1D nature of $p_{x'}$ and $p_{y'}$ electrons we discussed[20,21] (see Supplementary Note 1 for a tight-binding analysis). Various band crossings can be split and shifted when the magnetic moments of Ce$^{3+}$ order antiferromagnetically below 2.7 K or ferromagnetically in an external magnetic field[18]. In the paramagnetic phase above 2.7 K, signatures indicating proximate Kondo physics have been

observed[22,23]. Stoichiometric CeSbTe is difficult to synthesize, as Te atoms ($x$) tend to replace Sb atoms in the square lattice, or leave behind vacancies ($\delta$) in the CeTe layers, as represented by the formula $Ce(Sb_{1-x}Te_x)Te_{1-\delta}$. With increasing $x$, additional electrons are doped into the Sb square lattice. The Dirac node sinks below the Fermi energy ($E_F$) (Fig. 1h) and the Fermi surface arising from the Sb $5p$ orbitals splits into two concentric, diamond-shaped sheets (Fig. 1i). CDW states appear with $1 \times 5 \times 1$, $1 \times 3 \times 1$, and $3 \times 3 \times 2$ superstructures around $x$ values of 0.49, 0.66, and 0.89, respectively[19,23]. The increase of the CDW wave vector with increasing $x$ suggests that it is related to the nesting wave vector of the double-diamond Fermi surface, which also grows with increasing $x$ (arrows in Fig. 1i). X-ray diffraction has revealed that in these CDW phases, the Sb-Sb bond lengths show the greatest disproportionation and the Sb square lattice distorts into various bonding patterns[19]. The question remains as to how the electronic orbitals organize in these states.

Here, we use scanning tunneling microscopy (STM) to perform real-space imaging of a $1 \times 7$ CDW state in a $Ce(Sb_{1-x}Te_x)Te_{1-\delta}$ crystal with estimated $x$-0.39 and $\delta$-0.04 (see Supplementary Note 3 for an estimate of the Te vacancy concentration). As an atomically-resolved tool, STM enables us to elucidate the internal CDW structure with a high level of detail. Although the CDW originates predominantly from the Sb square lattice two atomic planes below the cleaved Te surface, by using an appropriate tunneling bias voltage where the Te density of states (DOS) is suppressed, we can image electronic contributions from the subsurface Ce $4d$ and Sb $5p$ orbitals. The constant-current topography reveals a complex state of atomic-sized lobes of charge density with spatial distribution spreading away from the Te positions and towards the Ce and Sb-Sb bond positions. The lobes are anisotropic and their axis of anisotropy varies periodically in space, alternating between the crystalline $a$ and $b$ axes. We interpret this pattern as a superposition of Sb $5p_x$ and $5p_y$ bond density waves that are identical in structure, but shifted in phase relative to each other. The $p_x$ and $p_y$ bond density waves are essentially building blocks whose superposition can give rise to CDWs with nontrivial orbital patterns. Our measurements demonstrate that $R$SbTe and related compounds are not only platforms for magnetic topological bands, as frequently discussed, but also hosts to spatially modulated $p$-electron phases.

## Results

### STM imaging of CDW at negative bias voltage

We performed STM imaging of cleaved single crystals of $Ce(Sb_{1-x}Te_x)Te_{1-\delta}$ at a temperature of 4.2 K and a magnetic field of 0 T, which lie in the paramagnetic phase of CeSbTe (see Supplementary Note 4 for the magnetic phase diagram). Figure 2a presents a typical STM topography acquired with a negative sample-tip bias voltage of −0.5 V, which involves tunneling from the filled orbitals of the sample. The image reveals a square lattice of isotropic lobes with an average distance of 4.4 Å, as extracted from the Fourier transform (FT) (Fig. 2b). The lattice constant immediately excludes the Sb square lattice with Sb-Sb distance of 3.08 Å as the origin of the observed lattice of bright lobes, but is consistent with either the Ce or Te sublattices of the CeTe buffer layer with Ce-Ce and Te-Te distances of 4.35 Å. Since the weakest bonds in the crystal lie between two adjacent Te planes (shaded plane in Fig. 1g), the surface termination is mostly likely a Te plane. This identification is corroborated by the sizeable number of vacancies ~4%, as expected for Te in $Ce(Sb_{1-x}Te_x)Te_{1-\delta}$ (see inset of Fig. 2a for a higher resolution image). We did not observe any other kinds of surface terminations.

In addition to the atomically resolved Te lattice, we spotted a unidirectional CDW running parallel to the Te-Te bond directions, i.e., the direction of the nesting wave vectors shown in Fig. 1i. In Fig. 2a, the CDW appears as a weak amplitude modulation superimposed on the atomic corrugations. The CDW is better visualized in an autocorrelation of the topography (Fig. 2c), which reveals an amplitude modulation of length $a$ that corresponds to the atomic lattice, plus a longer amplitude modulation of length ~$7a$ that corresponds to the CDW (see line cut in Fig. 2c). The CDW is also observable as superstructure peaks in the FT (Fig. 2b), from which we extract the actual CDW wavelength in the field of view to be $6.85a$, which is slightly smaller than $7a$, likely due to local discommensurations. By analyzing over 30 topographic images, we determine the average CDW wavelength to be $6.9 \pm 0.4a$ for the $Ce(Sb_{1-x}Te_x)Te_{1-\delta}$ sample presented here in the main text (see Supplementary Note 3 for statistics).

Figure 2d presents another negative-bias topography acquired over a larger field of view of ~$90 \times 200$ nm. We observed four domains, two with CDW along the horizontal axis of the image, and the other two with CDW along the vertical axis of the image. The domains range from roughly 30 to 100 nm in width. The 90° rotation of the CDW modulation can be directly seen, but is also confirmed by FTs of selected areas within single domains (Fig. 2e). The observation of rotated domains helps exclude tip artifacts as the source of the unidirectional $1 \times 7$ CDW. At the boundary between domains, the overlap region between orthogonal CDWs is roughly two to three CDW periods, as seen in the Fourier-filtered image of Fig. 2f. The CDW is relatively local in character and thereby reasonably robust against disorder.

### STM imaging of CDW internal structure at positive bias voltage

The STM topography appears drastically different when measured at positive sample-tip bias voltages, i.e., via tunneling into the empty orbitals of the sample. Figure 3a, b compare constant-current topographies simultaneously obtained over the same $20 \times 20$ nm area at −0.5 and +1.0 V (see Methods for explanation of Multi Pass imaging mode). Their corresponding FTs are shown in Fig. 3c, d. In the −0.5 V topography, the aforementioned $1 \times 7$ background modulation is barely visible by eye. In the +1.0 V topography, the $1 \times 7$ pattern becomes strongly manifested, but with a complicated internal structure, rather than just a simple modulation of amplitude. Further contrasts between −0.5 and +1.0 V imaging are highlighted by inspecting the enlarged views of the two topographies in Fig. 3e, f. The −0.5 V topography consists of atomic-sized lobes of charge density that are isotropic in shape (dashed circles in Fig. 3e). The positions of the lobes, which we identify by extracting the positions of all the local maxima in the topography (dots in Fig. 3e), form a mostly square lattice, with occasional vacancies and dislocations. On the other hand, the +1.0 V topography shows atomic-sized lobes that are anisotropic and elongated along either the $x$ or $y$ directions (red and blue dashed ellipses in Fig. 3f), which correspond to the crystallographic $b$ and $a$ axes. Their spatial distributions are shifted relative to the positions of the isotropic lobes seen at −0.5 V (gray dots in Fig. 3f) and also distorted from a regular square lattice. In the remainder of the paper, we focus on the analysis and interpretation of these unusual features in the +1.0 V image.

Insights as to why the +1.0 V image appears so distinct can be gleaned from density functional theory (DFT) calculations (see "Methods" for details of DFT calculations). The computed DOS reproduces the asymmetric V-shaped $dI/dV$ measured by STM (Fig. 3g). The DOS consists of Sb contributions that are nearly constant as a function of energy, Ce contributions that show a V-shaped depression centered at $E_F$, and Te contributions that are large below $E_F$, but nearly zero above $E_F$. The schematics in Fig. 3h depict isosurfaces of charge density for a Te-terminated slab, computed by integrating the local density of states (LDOS) within the appropriate energy range corresponding to −0.5 and +1.0 STM constant-current imaging. When imaging at −0.5 V, electrons tunnel from occupied surface Te $p_z$ orbitals into the tip, so the isotropic lobes arranged on a square lattice in Fig. 3e can be identified with Te atoms. The weakness of the observed CDW amplitude here may be expected, as the periodic lattice distortions in the CeTe buffer layer is an order of magnitude smaller than

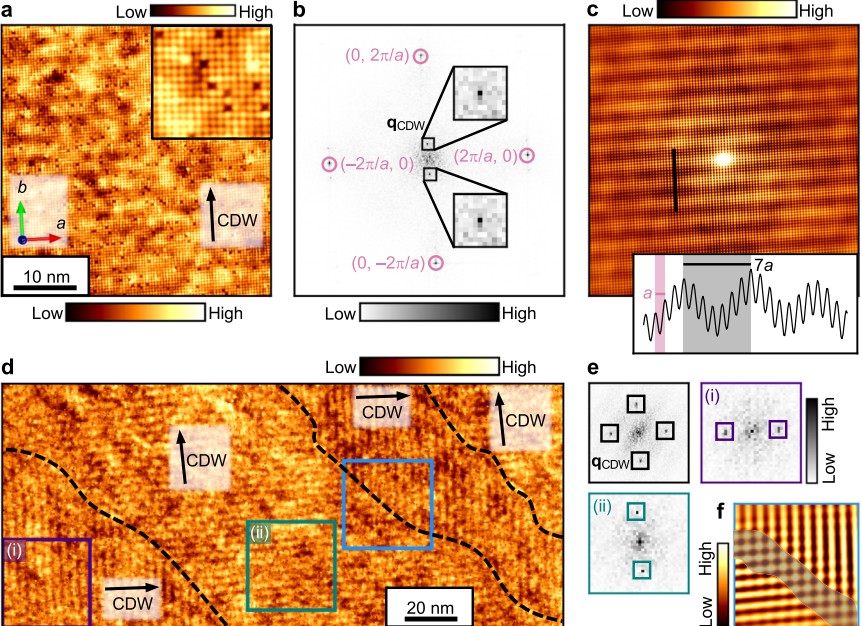

**Fig. 2 | STM imaging of unidirectional CDW on the Te-terminated surface.**
**a** Typical topographic image measured by STM with CDW direction marked by the black arrow. Inset: high-resolution topography of a smaller ~4 × 4 nm area, illustrating atomic resolution. Setpoint: −0.5 V, 80 pA. **b** Fourier transform (FT) of the area shown in (**a**). The Bragg peaks $q_{Bragg}$ and CDW peaks $q_{CDW} \approx 1/7 q_{Bragg}$ are enclosed in the pink circles and black squares, respectively. **c** Autocorrelation of (**a**). The inset shows a height profile corresponding to the black line cut.

**d** Topography of a larger field of view, revealing domains of orthogonal unidirectional CDW with directions marked by the black arrows. The domain boundaries are indicated by the dashed black lines. Setpoint: −0.5 V, 80 pA. **e** FTs of the entire field of view in (**d**) (upper left subpanel) and of restricted areas (i) and (ii) (upper right and lower left subpanels). **f** Blue boxed region in (**d**) with Fourier filtering around the CDW peaks, illustrating a narrow boundary region between domains where the CDW appears bidirectional (shaded region).

those of the CDW-active Sb layer[24]. When imaging at +1.0 V, electrons tunnel from the tip into unoccupied orbitals of the surface, but since there are nearly no available Te orbitals, the tunneling electrons have increased overlap with subsurface Ce $dx^2-y^2$ and Sb $p_x$ and $p_y$ orbitals. The anisotropic shape of the atomic-sized lobes in Fig. 3f suggests that they include contributions from these subsurface orbitals, and their spatial distributions are indeed shifted laterally from the Te atomic positions. Since the Sb orbitals are involved, the internal texture of the 1 × 7 CDW state becomes resolvable. We develop a detailed microscopic interpretation in the "Discussion" section.

**Elucidation of CDW internal structure**

The +1.0 V topography visibly reveals that the anisotropic lobes of charge density are organized into an intricate pattern. There appears to be a 1 × 7 periodic alternation between anisotropic lobes elongated along the $x$ direction and anisotropic lobes elongated along the $y$ direction, forming a ladder-like motif with horizontal "rungs" and vertical "rails." We performed a few steps of analyses to further elucidate the intrinsic underlying structure. First, we used the simultaneously acquired −0.5 V topography of Fig. 3a to construct a reference CeSbTe lattice. We applied Fourier filtering around the Bragg peaks (pink circles in Fig. 3c) to enhance the regularity of the square lattice (Fig. 4a). (See Supplementary Note 5 for the masks used in Fourier filtering.) The positions of the local maxima extracted from this filtered −0.5 V topography represent Te atomic positions (depicted as brown open circles in Fig. 4a, b), from which we could also interpolate the Sb-Sb bonds (gray lines in Fig. 4b) and Ce atomic positions (teal dots in Fig. 4b). Second, we extracted the positions of the local maxima of the anisotropic lobes in the +1.0 V topography (black dots in Fig. 4c) and placed their coordinates over the reference CeSbTe lattice (Fig. 4d). Although the anisotropic lobes are spatially extended objects, for the purpose of analysis, we indexed their positions according to their local maxima. Third, we characterized the $x$ or $y$ anisotropy of these lobes as follows: Within the topography $z(x, y)$, we computed and compared

the magnitudes of the second partial derivatives $\frac{\partial^2 z}{\partial x^2}$ and $\frac{\partial^2 z}{\partial y^2}$ evaluated at the local maxima $(x_i, y_i)$ of every anisotropic lobe. If $\left|\frac{\partial^2 z}{\partial x^2}(x_i, y_i)\right| < \left|\frac{\partial^2 z}{\partial y^2}(x_i, y_i)\right|$, then the local curvature of the topography is smaller along $x$ than along $y$, meaning that the anisotropic lobe is elongated more along $x$ than along $y$. If $\left|\frac{\partial^2 z}{\partial y^2}(x_i, y_i)\right| < \left|\frac{\partial^2 z}{\partial x^2}(x_i, y_i)\right|$, then the anisotropic lobe is elongated more along $y$ than along $x$. We quantified the degree of anisotropy as

$$\eta(x_i, y_i) = \left|\frac{\partial^2 z}{\partial x^2}(x_i, y_i)\right| - \left|\frac{\partial^2 z}{\partial y^2}(x_i, y_i)\right| - \left(\left|\frac{\partial^2 z}{\partial x^2}(x,y)\right|_{av} - \left|\frac{\partial^2 z}{\partial y^2}(x,y)\right|_{av}\right),$$
(1)

where $\left|\frac{\partial^2 z}{\partial x^2}(x_i, y_i)\right|$ is the magnitude of the second partial derivative evaluated at a lobe maximum $(x_i, y_i)$ and $\left|\frac{\partial^2 z}{\partial x^2}(x,y)\right|_{av}$ is the average of the magnitudes of the second partial derivatives computed for all pixels in the field of view. The subtraction of the constant term $\left(\left|\frac{\partial^2 z}{\partial x^2}(x,y)\right|_{av} - \left|\frac{\partial^2 z}{\partial y^2}(x,y)\right|_{av}\right)$ in Eq.1 accounts for any small global anisotropy that may originate from an imperfect tip. In Fig. 4d, the sign of $\eta(x_i, y_i)$ is represented by the orientation of the rectangles, horizontal for $\eta(x_i, y_i) > 0$ and vertical for $\eta(x_i, y_i) < 0$. The value of $\eta(x_i, y_i)$ is represented by the red-blue color scale, stronger red for increasing $x$ anisotropy and stronger blue for increasing $y$ anisotropy.

Figure 4e presents a composite image that encapsulates the underlying internal structure of the 1 × 7 CDW state seen in the +1.0 V topography. The anisotropic lobes are represented as horizontal red or vertical blue rectangles according to their $x$- or $y$-oriented anisotropy, and the reference CeSbTe unit cells are schematically included. The aforementioned ladder-like motif is clearly evident as periodically alternating striped regions of red and blue rectangles. Less obvious by eye in Fig. 4e is whether there is any pattern or trend for the positions of the red and blue rectangles. By computing 2D histograms in Fig. 4f of the positions of all the

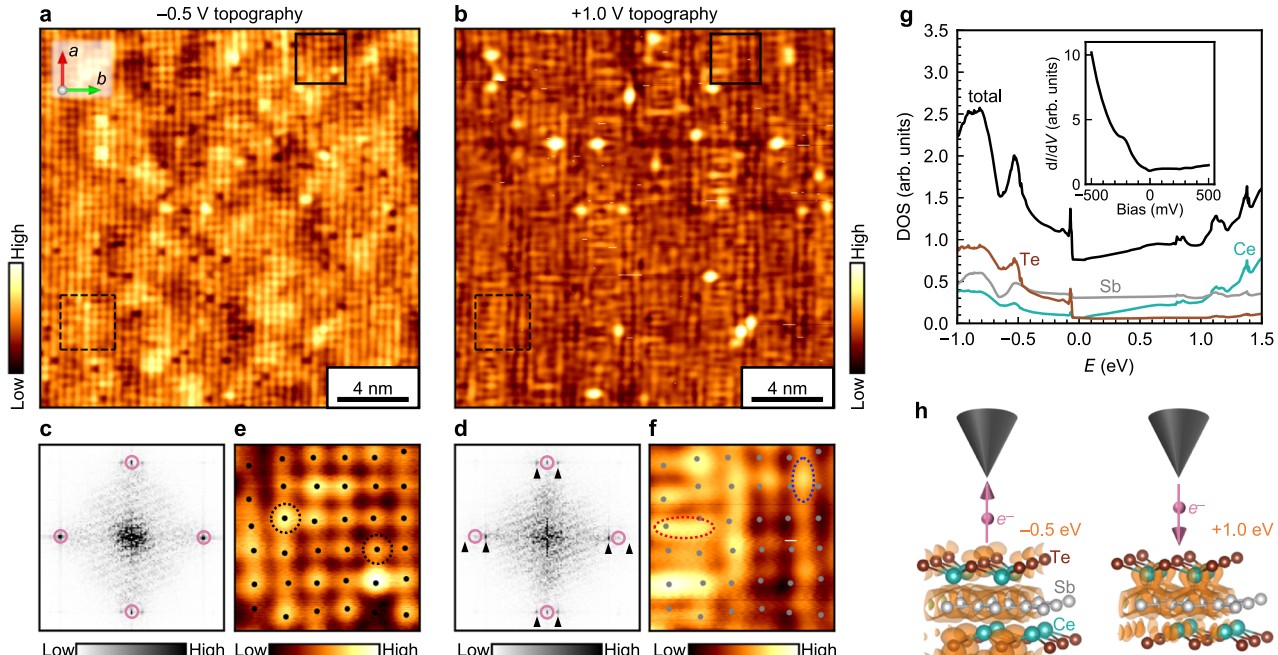

**Fig. 3 | Imaging CDW internal structure by varying bias voltage. a, b** STM topographies of the same ~20 × 20 nm field of view imaged with −0.5 and +1.0 V biases, respectively. Setpoint current: 20 pA. **c, d** FTs of (**a**) and (**b**), respectively. Bragg and CDW satellite peaks used for Fourier filtering are marked by pink circles and black triangles, respectively. **e, f** Enlarged view of the solid boxed regions in (**a**) and (**b**), respectively. The dashed circles and ellipses highlight the shapes of the charge density lobes. The black dots in (**e**) indicate points of local maxima in the topography. The gray dots in **f** are the positions of the local maxima in (**e**) superimposed. **g** DFT-computed total and atomically resolved DOS of CeSb$_{0.63}$Te$_{1.37}$. The inset is a $dI/dV$ spectrum measured via STM. Setup conditions: −0.5 V, 100 pA. Lock-in excitation: 20 mV. **h** DFT-computed isosurfaces of charge density, illustrating the orbitals involved in the tunneling processes of a Te-terminated surface. The charge density on the left is computed by integrating the LDOS between −0.5 and 0 eV (=$E_F$), matching the −0.5 V topography in (**a**), while the charge density on the right is computed by integrating the LDOS between 0 and +1.0 eV, matching the +1.0 V topography in (**b**).

anisotropic lobes, we reveal that the $x$- and $y$-oriented lobes exhibit distinct spatial distributions within the CeSbTe unit cell. The positions of the $x$-anisotropic lobes show large scatter along the $x$ axis, but are more confined along the $y$ axis. Their $y$ coordinates are clustered around the lower half of a Te-centered unit cell, just below the Te atom. Their distribution clearly breaks the mirror plane intersecting the Te atom along the $x$ axis. The positions of the $y$-anisotropic lobes show, on the contrary, larger scatter along the $y$ axis, but are more confined along the $x$ axis. Their $x$ coordinates are bimodally distributed towards the left and right sides of the Te-centered unit cell. (See Supplementary Note 6 for additional examples revealing the same internal CDW structure.)

The analyses that culminate in 4e, f capture the essential internal features of the 1 × 7 CDW state in CeSbTe. A sizable level of disorder, likely due to Te substitutions in the subsurface Sb layer, is nevertheless still present in these images. On one hand, STM is well-suited to resolve the local impact of defects on CDW. On the other hand, to enhance our visualization of the intrinsic CDW pattern, we additionally performed Fourier filtering of the +1.0 V topography with a twofold symmetric mask that includes the Bragg peaks and first harmonics of the CDW satellites (pink circles and black triangles in Fig. 3d). The resulting topography in Fig. 5a manifests the ladder-like motif much more clearly. Three periods of such "ladders" are shown in Fig. 5b. Repeating the same analyses of Fig. 4 with this filtered topography, we find the scatter in the positions of the $x$- and $y$-anisotropic lobes within the CeSbTe unit cell to be much reduced. The coordinates of the $x$-anisotropic lobes spread somewhere between the Te atom and the two Sb-Sb bonds in the lower half of the Te-centered unit cell. The coordinates of the $y$-anisotropic lobes show a sharper bimodal distribution towards the left and right sides of the Te-centered unit cell compared to that in

Fig. 4f and appear to cluster around Sb-Sb bonds and the Ce atoms. This spatial arrangement of anisotropic lobes is also visible locally within a CDW period, as seen in Fig. 5e. In Fig. 5d, we binned the filtered +1.0 V topography into red or blue regions according to the predominant $x$ or $y$ orientation of the anisotropic lobes. Figure 5e shows the enlarged view of a small region. The anisotropic lobes look like they partially overlap with Sb-Sb bonds forming zigzag patterns that extend either along the crystallographic $a$ or $b$ directions, which is reminiscent of the pictures presented in Fig. 1f.

## Discussion
### Microscopic interpretation
Here, we develop our microscopic interpretation of the +1.0 V STM topography. At positive energies, the DOS from the surface Te atoms is largely suppressed (Fig. 3h). The anisotropic lobes are therefore likely a convolution of remnant Te states with subsurface Ce and Sb states, as the local maxima of the lobes do not perfectly coincide with any of the Te, Ce, and Sb atomic positions (Figs. 4f and 5c). We suggest that the anisotropy of the lobes is inherited from underlying zigzag chains formed by Sb $p_x$ and $p_y$ orbitals, which are then filtered through Ce and remnant Te states. To verify our microscopic interpretation, we performed DFT slab calculations of the simpler 1 × 1 CDW state of CeSbTe and computed the integrated LDOS between 0 and +1.0 eV, i.e.,

$$\rho_{int}(x, y, z, +1.0\,eV) = \int_{0\,eV}^{+1.0\,eV} N(x, y, z, E)\,dE, \quad (2)$$

where $N(x, y, z, E)$ is the LDOS derived from the Kohn-Sham orbitals. As $\rho_{int}$ is a function of three variables ($x$, $y$, $z$), we visualize this quantity in 3D by plotting iso-amplitude surfaces, i.e., $\rho_{int}(x, y, z, +1.0\,eV) = \rho_0$. Three such isosurfaces are shown in Fig. 6a, with

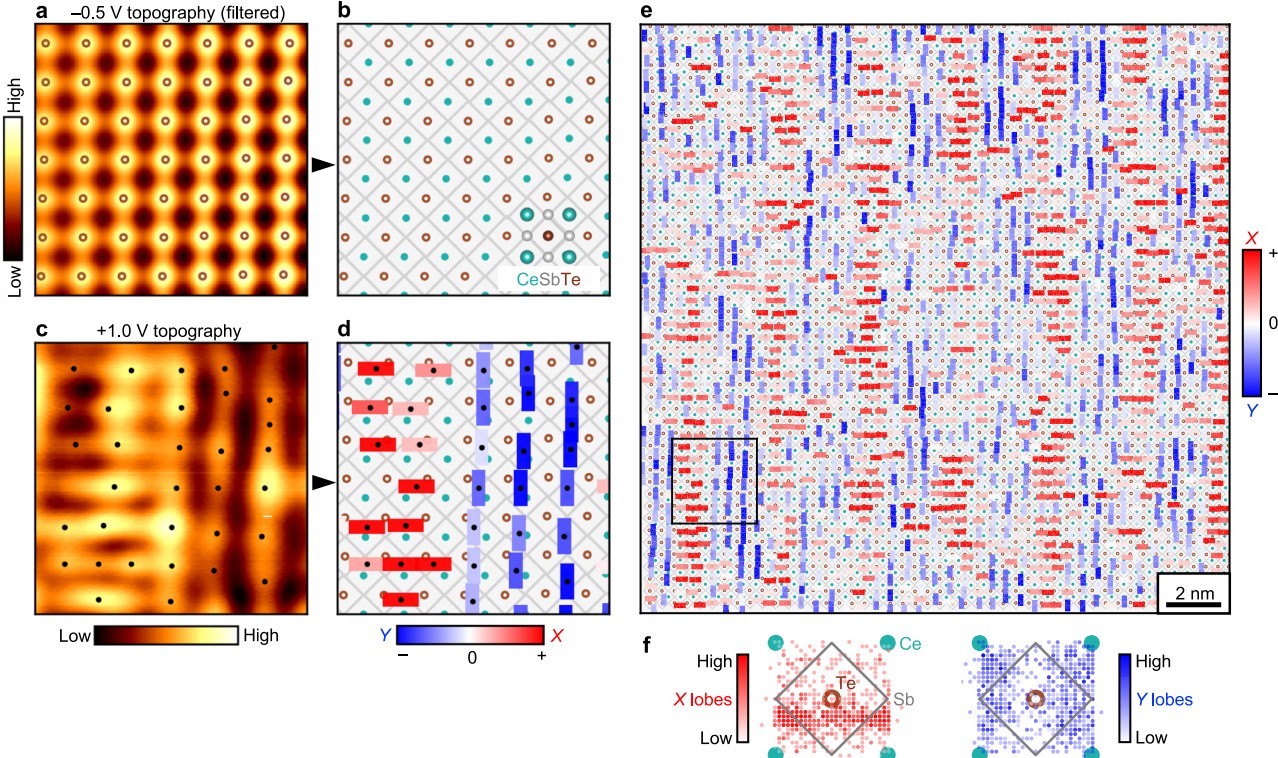

**Fig. 4 | Visualizing the internal structure of CDW (I): map of anisotropic lobes.** **a** Selected region of the −0.5 V topography of Fig. 3a with Fourier filtering around the Bragg peaks. The brown open circles indicate the positions of the local maxima, i.e., the Te atomic positions. **b** Reference CeSbTe lattice extracted from (**a**), with Ce represented as teal dots and Sb-Sb bonds as diagonal gray lines. **c** Same region as (**a**), extracted from the +1.0 V topography of Fig. 3b. The black dots indicate the positions of the local maxima. **d** Overlay of anisotropic lobes on the reference CeSbTe lattice. The anisotropic lobes are schematically depicted as rectangles with orientation along $x$ or $y$. Their colors indicate the degree of lobe anisotropy, as quantified by $\eta(x_i, y_i)$ in Eq. 1. **e** Map of lobe anisotropy for the entire field of view of Fig. 3b. The black square encloses the selected region shown in (**a**–**d**). (This region is also indicated in Figs. 3a, b by the dashed black squares.) **f** Distribution of the positions of the $x$- and $y$-anisotropic lobes from (**e**) within a reference CeSbTe unit cell. The intensity of the red or blue color is proportional to the number of anisotropic lobes at the given position.

decreasing values of $\rho_0 = 2.5 \times 10^{-3}$, $9.5 \times 10^{-4}$, and $5 \times 10^{-4}$ arb. units from left to right. The regions of maximum charge density (largest $\rho_0$) are concentrated in the Sb plane and appear as zigzag lobes that resemble the $p_{x'}$ and $p_{y'}$ dimer chains discussed for the simple square lattice model (Fig. 1f). The zigzag lobes here are directed away from the stronger Sb-Sb bonds and overlap along the weaker Sb-Sb bonds (dashed black ellipses in Fig. 6a); i.e., they are antibonding in nature. At positive biases as large as +1.0 V, it is natural that we probe the antibonding states associated with the CDW. As $\rho_0$ decreases and the charge density spreads away from the Sb plane and towards the Te surface, the hybridization of the antibonding Sb $p_{x'}$ and $p_{y'}$ orbitals with Ce $dx^2–y^2$ orbitals, which have their lobes directed along $x$ or $y$ (the crystallographic axes), likely serves to reshape the zigzag lobes of charge density into more rounded lobes elongated along $x$ or $y$ (dashed blue ellipses in Fig. 6a). When the charge density spreads to the Te surface at low values of $\rho_0$, we no longer resolve the fine details of individual Sb $p_{x'}$ and $p_{y'}$ orbitals, but their zigzag extension along $x$ or $y$ is the microscopic origin of the $x$ and $y$ anisotropy of the lobes imaged at +1.0 V. (See Supplementary Note 7 for simulations of STM topography in the 1 × 1 CDW state with anisotropic lobes).

Based on our microscopic interpretation of the anisotropic lobes, we extract the underlying texture of the Sb $p$ orbitals. The local maxima of the anisotropic lobes imaged at +1.0 V should be located close to the weaker Sb-Sb bonds, while the local minima between the lobes should be located close to the stronger Sb-Sb bonds. In the field of view of Fig. 6b, we extract all the Sb-Sb bonds which lie in the vicinity of a local minimum (white regions) between

the anisotropic lobes (red and blue regions). These bonds together constitute the underlying pattern of the CDW in the Sb layer, which consists of zigzag chains that alternate along the $x$ and $y$ directions.

Figure 6c–e show a decomposition of the experimentally determined 1 × 7 CDW structure into a superposition of $p_{x'}$ and $p_{y'}$ bond density waves. While the patterns are highly nontrivial, a closer inspection reveals a fundamental similarity to the $p_{x'}$ and $p_{y'}$ bond density waves of the simpler 1 × 1 CDW (Fig. 1e). The patterns in Fig. 6d, e comprise chains of $p_{x'}$ and $p_{y'}$ orbitals along the $x'$ and $y'$ directions, respectively, that are almost regularly dimerized, and the neighboring chains have a relative phase shift of $a'$. Along each chain, however, there are periodic phase slips, such as the occurrence of two consecutive strong bonds, i.e., a trimer, or two consecutive weak bonds (examples circled in Fig. 6d). These phase slips effectively enlarge the supercell from 1 × 1 to 1 × 7 and distinguish the 1 × 7 CDW from the 1 × 1 CDW.

The presence of phase slips unveils the role of electron doping at a microscopic level. In stoichiometric CeSbTe, Sb holds a formal oxidation state of −1, which results in nearly half-filled $p_{x'}$ and $p_{y'}$ orbitals on the Sb square lattice, as described by Fig. 1. In non-stoichiometric Ce(Sb$_{1-x}$Te$_x$)Te, the additional electrons donated by Te occupy the antibonding states of Sb$^{1-}$, which has a destabilizing effect on the square lattice[25–27]. In reciprocal space, the Fermi surface splits into two concentric, diamond-shaped pockets, which changes the nesting conditions to favor a larger CDW periodicity. (See Supplementary Note 8 for a further discussion of nesting wave vectors.) In real space, the electron doping can be simply

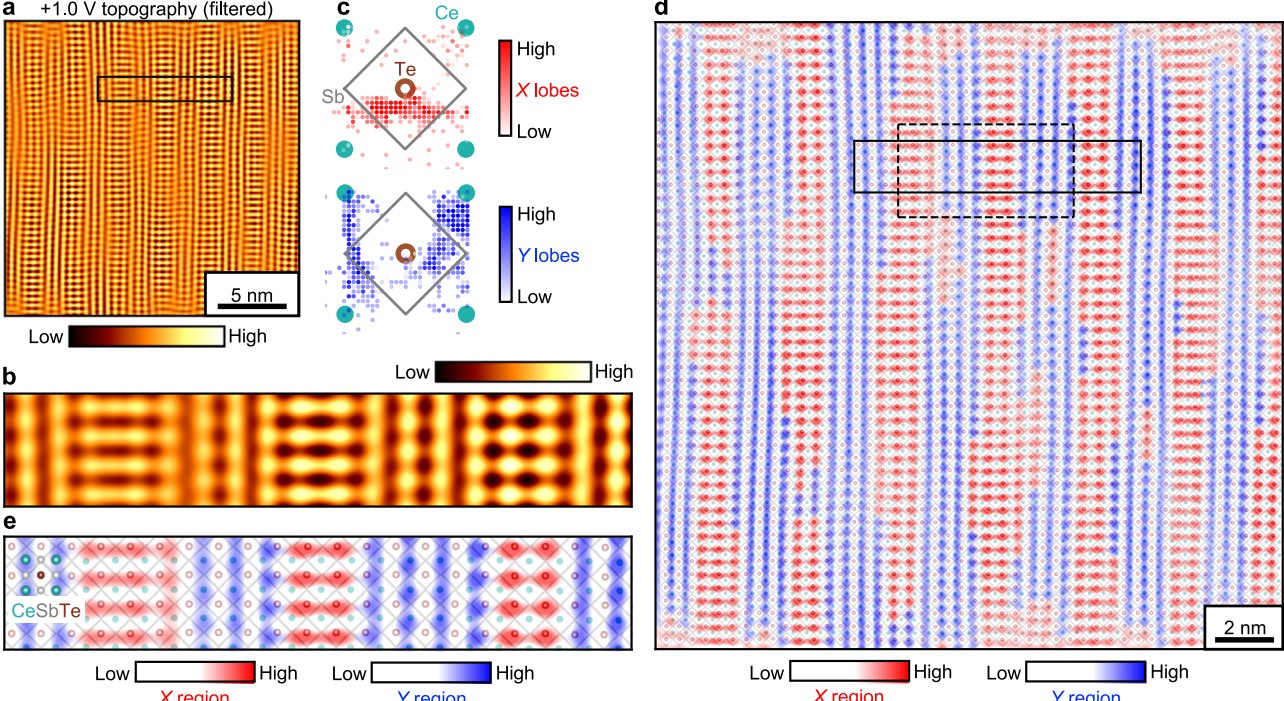

**Fig. 5 | Visualizing the internal structure of CDW (II): Fourier-filtered pattern.** **a** +1.0 V topography of Fig. 3b with Fourier filtering around the Bragg and CDW satellite peaks. **b** Enlarged view of the boxed region in (**a**). **c** Distribution of the positions of the *x*- and *y*-anisotropic lobes within a reference CeSbTe unit cell after applying the Fourier filtering in (**a**). **d** Filtered topography of (**a**) with red-blue binning applied to demarcate regions of predominant *x* (red) or *y* (blue) anisotropy. The Ce, Sb, and Te atomic positions are included. **e** Enlarged view of the solid boxed region in (**d**).

understood as weakening the original dimerized bonds of the $1 \times 1$ CDW and creating periodic phase slips, such that the enlarged periodicity matches the new nesting conditions. Our microscopic interpretation thus bridges the reciprocal-space picture of nesting, which determines the preferred CDW periodicity, with the real-space picture of dimerization in covalently bonded networks of *p* electrons.

We arrive at a common microscopic mechanism for the patterns of the $1 \times 1$ and $1 \times 7$ CDW. Simultaneous nesting of quasi-1D $p_{x'}$ and $p_{y'}$ Fermi sheets in reciprocal space leads to a superposition of $p_{x'}$ and $p_{y'}$ bond density waves in real space. There is a phase degree of freedom in how these two density waves should be superimposed, which ultimately is fixed by higher-order hopping terms and other additional interactions that hybridize neighboring and intersecting chains of $p_{x'}$ and $p_{y'}$ dimers. In the case of the $1 \times 1$ CDW, due to the regular structure of the $p_{x'}$ and $p_{y'}$ dimer chains, we could only obtain a simple pattern of zigzag bonds that uniformly extend along *x* or *y* (Fig. 1f). In the case of the $1 \times 7$ CDW, the phase slips in the $p_{x'}$ and $p_{y'}$ dimer chains enable more complicated orbital textures to emerge, such as the pattern of alternating *x*- and *y*-oriented zigzag bonds that we observed. The bulk $1 \times 3$ and $1 \times 5$ CDW states of CeSbTe[19] can be similarly understood in this way (see Supplementary Note 9 for comparison with x-ray diffraction data). The CDW in *R*Te₃ plausibly also involves some superposition of $p_{x'}$ and $p_{y'}$ bond density waves resulting in a nontrivial orbital texture and an amplitude mode with axial symmetry[13,15,16].

We note that our microscopic picture considers only the dominant role of Sb 5*p* orbitals. In reality, the anisotropic lobes show additional complexity, including an inequivalent shift of the *x*-oriented lobes towards the Te atoms and the *y*-oriented lobes towards the Ce atoms. The CeTe buffer layer likely participates in the CDW state, albeit with much smaller periodic lattice distortions[24], and the hybridization between Sb *p* and Ce *d* orbitals may play a role in determining the

actual pattern of the Sb *p*-orbital bond density wave. The role of Ce and Te atoms remains to be further clarified.

## Outlook

The orbital texture of the CDW on a *p*-electron square lattice is a consequence of the $p_{x'}$ and $p_{y'}$ orbital degree of freedom inherent to the model. Despite the apparent simplicity of this mechanism, its significance has not been appreciated in *R*Te₃ until recently[13,15,16] and directly imaged in *R*SbTe until now. We note that similar concepts have been discussed in other material systems harboring multi-*q* CDW, where the CDW state is composed of multiple wave vectors connecting distinct orbital sectors of the Fermi surface[28–30]. For example, the $2 \times 2 \times 2$ CDW in 1*T*-TiSe₂ is composed of three transverse, single-*q* modes with distinct combinations of Ti 3*d* and Se 4*p* orbitals[31]. There is a degree of freedom associated with the relative weights and phases with which the three transverse modes should be summed in real space, which could result in nontrivial orbital textures with chiral symmetry.

We also remark that a similar charge-order pattern of alternating anisotropic lobes has been famously imaged in STM experiments of cuprate superconductors[32–35]. There, the charge-order period is around 4*a* and orthogonal domains coexist on the scale of several nanometers, giving rise to a charge order that appears more bidirectional ("checkerboard") in nature. Nevertheless, the anisotropic lobes are spatially arranged with the same ladder-like motif. They are ascribed to O 2$p_x$ and 2$p_y$ orbitals in the CuO₂ plane that modulate out of phase with each other, endowing the charge order with a *d*-symmetric form factor[34–36]. There are some fundamental differences, however, between the cuprates and *R*SbTe. In the cuprates, the degeneracy of the $p_x$ and $p_y$ orbitals is already lifted at each O site due to its local twofold symmetry, being bonded to two neighboring Cu atoms. The modulating pattern of O 2$p_x$ and 2$p_y$ orbitals is also likely driven by their hybridization to localized Cu

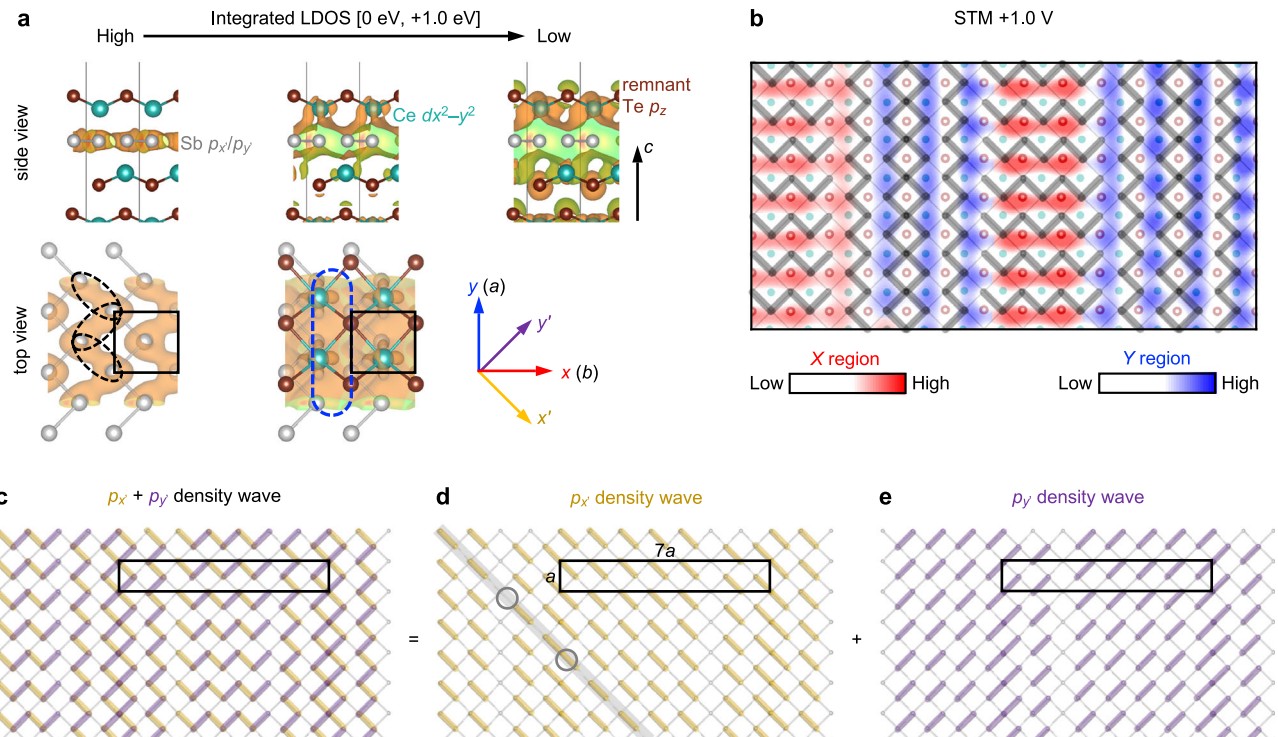

**Fig. 6 | Bond density wave of Sb $5p_{x'}$ and $5p_{y'}$ orbitals as microscopic origin.**
**a** Integrated LDOS between 0 and +1.0 eV of a 1 × 1 CDW state in a CeSbTe slab computed by DFT. To visualize the integrated LDOS, three iso-amplitude surfaces of charge density are shown, i.e., $\rho_{int}(x, y, z, +1.0\,eV) = \rho_0$, as described by Eq. 2. The value of $\rho_0$ decreases from left to right. The first two isosurfaces are shown in two perspectives. **b** Filtered STM topography with red-blue binning, reproduced from the dashed boxed region of Fig. 5d. The inferred positions of the stronger Sb-Sb bonds are highlighted with thick gray lines. **c** Bond density wave pattern of $p_{x'}$ and $p_{y'}$ orbitals with a 1 × 7 supercell (black rectangle). **d**, **e** Decomposition of (**c**) into individual $p_{x'}$ and $p_{y'}$ bond density waves, respectively. The patterns are similar to those of Fig. 1e, except for periodic phase slips (gray circles) along the dimer chains (gray line) that effectively enlarge the CDW supercell to 1 × 7.

$3dx^2-y^2$ orbitals with strong electronic correlations[37]. In $R$SbTe, the degeneracy of the $p_{x'}$ and $p_{y'}$ orbitals is preserved at each Sb site, which is bonded to four neighboring Sb atoms. The Sb square lattice here also represents a purer $p$-electron system, where the orbital texture is likely determined by interactions involving only $p_{x'}$ and $p_{y'}$ orbitals. It remains to be clarified whether the striking similarities between the CDW of $R$SbTe and the charge order of cuprates are simply coincidental, or reflect some generic microscopic physics of $p$ orbitals.

In conclusion, we have utilized STM to reveal the internal structure of the 1 × 7 CDW state of nonstoichiometric $Ce(Sb_{1-x}Te_x)Te_{1-\delta}$. As future steps, it would be interesting to investigate the evolution of this texture as the CDW wavelength changes across a wider doping range $x$[38], or the effects of uniaxial strain. In addition, it would be interesting to clarify whether this CDW has hidden orbital angular momentum, similar to that suggested by Raman spectroscopy for $GdTe_3$ and $LaTe_3$[13], and if so, whether the orbital angular momentum and resulting CDW pattern can be modified by flipping the $Ce^{3+}$ moments with an external magnetic field[39].

## Methods

### Experimental

Single crystals of $Ce(Sb_{1-x}Te_x)Te_{1-\delta}$ were grown with iodine vapor transport[18]. Magnetotransport measurements were performed with a physical property measurement system from Quantum Design using the standard four-probe method. The magnetic field was applied along the $c$ axis while the current flowed in the $a$-$b$ plane. We conducted STM experiments using a home-built, low-temperature, ultrahigh vacuum (UHV) instrument. The sample was cleaved in UHV on a cleaving stage cooled by liquid nitrogen, then quickly transferred to the microscope head held at 4.2 K. The topographic

images were acquired in the constant-current mode. Unless otherwise indicated, the only processing performed on the topographic images shown in the manuscript (Figs. 2a, d, 3a, b, e, f, and 4c) were a background plane subtraction, or a row-by-row parabolic subtraction. Differential conductance spectra were obtained with the standard lock-in amplifier technique. We used a root-mean-square oscillation voltage between 1.4 and 20 meV and a lock-in frequency of 973 Hz.

To obtain the topographies at −0.5 and +1.0 V in Fig. 3a, b over the same area, with each pixel matched at the same position, we used the Multi Pass module of the Nanonis SPM Controller software. The Multi Pass mode enables each line of an image to be scanned multiple times using different parameters. Using a current setpoint of 20 pA, we scanned each line three times, with bias voltages of −0.5, +1.0, and −0.5 V, respectively. The first two images are shown in Figs. 3a, b, while the third image served as a control to ensure minimal drift during the repeated scans of each line (see Supplementary Note 10).

### Data analysis
We describe our analysis algorithms in greater detail:

**Detecting positions of local maxima.**
1. Image parameters: ~20 × 20 nm, 1216 × 1216 pixels.
2. Pre-processing (only for the purpose of detecting local maxima): For the topographic images (Fig. 3b), we applied 2D Gaussian smoothing of width σ (=3 pixels) and computed the 2D Laplacian to aid the detection of local maxima. For the Fourier-filtered images (Figs. 4a and 5a), the images were smooth enough to skip this step.
3. We defined a grid over the image with spacing $n$ (=2 pixels). At every intersection point of the grid, we defined an $m \times m$ window

($m = 16$ pixels) centered about this point and extracted the coordinates of the image maximum within this window. This procedure could identify all the local maxima corresponding to the atomic-sized lobes, but greatly overcounted the number of actual local maxima.

4. We removed every local maximum that lies within a threshold distance $t$ (=16 pixels) of another local maximum. This removed duplicate points.

5. We removed every local maximum that lies on the boundary of the image.

### Computing lobe anisotropy.

1. Pre-processing (only for the purpose of computing lobe anisotropy): we applied 2D Gaussian smoothing of width $\sigma$ (=3 pixels) to the topographic image in Fig. 3b.

2. We computed the second partial $x$ and $y$ derivatives at every pixel of the image.

3. At the coordinates $(x_i, y_i)$ of every local maximum, we computed the anisotropy $\eta(x_i, y_i)$ as defined in Eq. 1

### Distribution of positions of anisotropic lobes within CeSbTe unit cell.

1. We identified the set of coordinates $\{(x_i^-, y_i^-)\}$ of the local maxima in the Fourier-filtered $-0.5$ V topography, i.e., the positions of the Te atoms.

2. We identified the set of coordinates $\{(x_i^+, y_i^+)\}$ of the local maxima in the (Fourier-filtered) $+1.0$ V topography, i.e., the positions of the $x$- and $y$-anisotropic lobes.

3. For each anisotropic lobe at $(x_i^+, y_i^+)$, we identified the Te atomic position $(x_j^-, y_j^-)$ that lies closest to $(x_i^+, y_i^+)$.

4. We computed the difference vector $(\Delta x_i, \Delta y_i) = (x_i^+ - x_j^-, y_i^+ - y_j^-)$.

5. We constructed 2D histograms of the difference vectors $(\Delta x_i, \Delta y_i)$, which represent the distribution of anisotropic lobes within a Te-centered unit cell (Figs. 4f and 5c).

### Red-blue binning of topography.

1. For each pixel of the topography in Fig. 5a, we determined the nearest anisotropic lobe and classified the pixel with an $x$ or $y$ label according to the $x$ or $y$ orientation of that nearest anisotropic lobe.

2. For all the pixels labeled "$x$," the topography is displayed according to a red-white color map, red being maximum and white being minimum.

3. For all the pixels labeled "$y$," the topography is displayed according to a blue-white color map, blue being maximum and white being minimum.

### DFT

We performed DFT calculations using the Vienna Ab initio Simulation Package (VASP)[40]. Since our STM measurements probed the paramagnetic phase of CeSbTe, we performed nonmagnetic calculations. We used the Perdew-Burke-Ernzerhof (PBE) parametrization of the generalized gradient approximation functional[41]. We treated the Ce $5s^2 5p^6 6s^2 5d^1$, Sb $5s^2 5p^3$, and Te $5s^2 5p^4$ electrons as valence. The occupied Ce $4f^1$ electron was frozen in the core, since angle-resolved photoemission spectroscopy detected the $4f$ band to lie 3.1 eV below the $E_F$[18], which is beyond the energy range of our STM measurements. The energy cutoff for the plane-wave basis was 450 eV.

We computed the band structure (Fig. 1h), Fermi surface (Fig. 1i), and DOS (Fig. 3g) of $Ce(Sb_{1-x}Te_x)Te$ with $x = 0.37$ as follows: Using the virtual crystal approximation, we replaced Sb atoms in the Sb square lattice with fictitious atoms whose pseudopotential was a weighted average of 63% of the Sb pseudopotential and 37% of the Te pseudopotential[42]. We adopted an orthorhombic $1 \times 1 \times 1$ unit cell with experimental lattice constants refined from powder x-ray

diffraction[19]: $a = 4.38895(6)$ Å, $b = 4.43760(7)$ Å, and $c = 9.33984(10)$ Å. The internal atomic coordinates were determined by structural optimization with a force tolerance of 5 meV/Å. We computed the DOS using a Brillouin zone sampling of $30 \times 30 \times 14$. For the Fermi surface, a different sampling of $32 \times 32 \times 16$ was used. For the band structure, spin-orbit coupling was incorporated in a non-self-consistent cycle.

For the charge densities shown in Fig. 3h, we constructed a $1 \times 1 \times 5$ slab of $CeSb_{0.63}Te_{1.37}$ with both surfaces terminated by a Te plane, again using the virtual crystal approximation to simulate the doping. We fixed the in-plane lattice constants to the same values of $a = 4.38895(6)$ Å and $b = 4.43760(7)$ Å, and used a $c$-axis size of 60 Å, which yielded a vacuum spacing of ~15 Å between periodic slabs. Internal atomic coordinates were relaxed with $12 \times 12 \times 1$ Brillouin zone sampling and a force tolerance of 10 meV/Å. A dipole correction was included to negate spurious interactions due to the slab asymmetry. After structural optimization, we calculated the wave function self-consistently with a Brillouin zone sampling of $44 \times 44 \times 1$. From the wave function, we then computed the LDOS, $N(x, y, z, E)$. The left schematic of Fig. 3h represents an isosurface of the integrated LDOS

$$\rho_{\text{int}}(x, y, z, -0.5\text{eV}) = \int_{-0.5\,\text{eV}}^{0\,\text{eV}} N(x, y, z, E)dE, \qquad (3)$$

corresponding to $-0.5$ V STM imaging. The right schematic of Fig. 3h represents an isosurface of the integrated LDOS $\rho_{\text{int}}(x, y, z, +1.0$ eV$)$, as defined in Eq. 2 which corresponds to $+1.0$ V STM imaging.

For the simulation of the $1 \times 1$ CDW state shown in Fig. 6a, we took as inspiration a $1 \times 1 \times 2$ CDW state proposed by Wang et al.[17] for nominally stoichiometric CeSbTe. In their proposed structure, each Sb layer form zigzag chains that uniformly extend along the $a$ axis, and the zigzag chains in adjacent Sb layers are shifted along the $b$ axis by half a unit cell. We constructed such a $1 \times 1 \times 2$ supercell of stoichiometric CeSbTe and determined the optimized lattice parameters by DFT, $a = 4.4006$ Å and $b = 4.4085$ Å. We then extended the supercell into a $1 \times 1 \times 5$ slab with a $c$-axis size of 60 Å, which yielded a vacuum spacing of ~15 Å between periodic slabs. By relaxing the internal atomic coordinates within VASP, we found that the Sb square lattices spontaneously formed zigzag chains along the $a$ axis with phase shift of $b/2$ across adjacent Sb layers. We could reproduce the CDW pattern described by Wang et al. with an average Sb-Sb bond disproportionation of ~8%, close to the experimental value of 7%.

We visualized crystal structures using VESTA[43] and Fermi surfaces using XCrySDen[44].

## Data availability

The data that support the findings in the main text are available through Figshare (https://doi.org/10.6084/m9.figshare.27255576). Extra raw data presented in the Supplementary Information are available from the corresponding author upon request.

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

## Acknowledgements
This project was funded by the Deutsche Forschungsgemeinschaft (DFG, German Research Foundation)—TRR 360— 492547816 (D.H. and H.T.). Sample growth was supported by the Gordon and Betty Moore Foundation's EPIQS initiative (grant number GBMF9064; L.S.), by an NSF CAREER grant (DMR-2144295; L.S.), and by the Air Force Office of Scientific Research under award number FA9550-20-1-0246 (L.S.). We thank M. Dueller and K. Pflaum for technical support and T.T.M. Palstra and A. Yaresko for useful discussions.

## Author contributions
X.Q. and Q.H. performed the STM experiments under the supervision of L.Z. S.L. and L.S. synthesized the samples. X.Q., D.H., and H.T. analyzed and interpreted the data. D.H. performed the DFT calculations. X.Q. and D.H. prepared the manuscript with input from all co-authors.

## Funding

## Competing interests

The authors declare no competing interests.
