## [Transparent Peer Review file · Nature Communications]

Visualizing the internal structure of the charge-density-wave state in CeSbTe

Corresponding Author: Dr Dennis Huang

Version 0:

Reviewer comments:

Reviewer #1

(Remarks to the Author)

In this manuscript, X. Que et al. reported an STM study on the CDW state of $\text{Ce}(\text{Sb}_{1-x}\text{Te}_x)\text{Te}_{1-\delta}$. They observed the anisotropic lobes of the charge density wave in the STM topography at positive bias dominated by Sb p states, while these anisotropic lobes were not observed at negative bias dominated by Te states. Then, in their px/py bond model in the Sb layers, they attributed these anisotropic lobes to zigzag chains of Sb-Sb bonds. Thus, the authors demonstrate the p-orbital texture in the 1x7 CDW of $\text{Ce}(\text{Sb}_{1-x}\text{Te}_x)\text{Te}_{1-\delta}$. I think the result is interesting, but some ambiguities need to be clarified. I can potentially suggest publication when the issues below are addressed:

1. The lattice constant of the Te termination measured in this study is 4.14Å, but the Te-Te distances should be 4.35Å. Though, due to the thermal drift, STM is not as precise as XRD in measuring the lattice constant, 5% is still a quite big error. I would suggest the authors check the piezo calibration. Or, if the 4.14Å were true, the optical measurement would be required to confirm this.
2. Fig. 3h shows the electrons tunnel to the deep Ce d orbitals and, then, deeper Sb p orbitals at positive bias. And in the rest of the work, the authors only focus on discussing Sb p orbitals. Usually, it is different to tunnel to the local d orbitals, but, in this case, Sb p orbitals are deeper than Ce d orbitals. I think the authors cannot ignore the contribution of Ce d orbitals in this work.
3. Considering the authors propose the inter CDW structure found at 1V is p orbital texture, it would be better to clarify the orbital texture in the dI/dV mapping versus energy rather than the topography. There may be bias-dependent effects or other effects shown in the data. For example, the red-blue ratio in fig. 4e(1V), S8i(1V), S9c(0.5V) and S10b(0.5V) may be different as visualized by my eyes. The red/blue represents the different anisotropic lobes.

Reviewer #2

(Remarks to the Author)

This article describes the observation of a modulated pattern of charge density in (slightly off-stoichiometric) CeSbTe. The authors interpret their findings in terms of two bond density waves in distinct orbital sectors, with a modulated pattern of relative phase shifts.

The experimental findings are timely, relevant to the field, and look interesting.

However, the following issues prevent me from recommending this work for publication in Nature Communications:

1) Context.

Although perhaps not crucial to the results themselves, I would like to point out that the state of the art is misrepresented in the introduction.

The authors claim that orbital-structured density waves have not received much attention in the literature yet, but they seem to have missed a whole section of the literature that is directly relevant to the results presented here. As a pointer for where to start looking, I would like to mention at least:

- Fukutome, Prog. Theor. Phys. 71, 1 (1984) -- this introduces orbital-structured density waves in p-orbital materials, precisely along the lines used in the present work. There is more work along these lines by the same author, and the orbital-structured density wave in these same p-orbital materials attracted attention from others as well.
- Whangbo & Canadell, J. Am. Chem. Soc. 114, 9587 (1992) -- lays the theoretical foundation for the general treatment of multiple-Q density waves in distinct orbital sectors. Again, there is more work by the same authors.
- Peng et al., Phys. Rev. Research 4, 033053 (2022) -- a recent example of experimental identification of the same type of orbital-structured charge density wave as the one discussed by the authors in the present work.

Again, these are just three examples of what is an active field of research, whose relevance to the current work ought to be explored by the authors.

Secondly, I wonder if the 1x7 density wave reported by the authors has been seen before? The authors mention 1x5, 1x3, and 3x3, but not 1x7.

If not, then why did the authors study this particular density wave and not one of the other known ones?

2) Experimental procedure.

The authors take great care to describe their experimental procedures and data analysis in detail, which is much appreciated.

However, I do have some major worries about the procedure used:

- The authors used a constant-current mode to scan the surface and map the charge density modulations. However, as recently pointed out in arxiv:2406.03294, constant-current mode is not generally suitable for imaging energy-dependent charge modulation patterns in charge density wave materials.

- On top of that, the authors discuss in detail the charge modulations obtained at +1V when tunneling into the third atomic layer, but they also show that already at -0.5V there is a period-7 modulation associated with the top atomic layer. As the tunneling current depends exponentially on distance, it is not clear to me how the authors can be certain that the period-7 pattern they observe at +1V can really be attributed to the Sb atoms in the third layer. Even if the Te DOS at +1V is very low, it is much closer to the tip. In constant-current mode, it is then not inconceivable that the spatial modulation in even a very small Te DOS can cause the tip-sample distance to vary and thereby impose a period-7 modulation on whatever other features come from the underlying atomic layers.

- According to fig 3g, the DOS of Ce atoms is as large as that of Sb atoms at +1V, but the Ce atoms are in the second atomic layer and the Sb atoms in the third. Why should the STM images at 1V not be dominated by Ce?

- Looking at the FT views in figure 3 as well as the Supplemental Material, the distance between Bragg peaks in the vertical direction seems larger than that in the horizontal direction. This would suggest that the crystal is strained, and has reduced rotational symmetry. It is then not clear whether the modulations observed by the authors are intrinsic to the material, or caused by the imposed strain.

- Finally, both in figure 3 and throughout the Supplemental Material, the peaks to the left and right seem to be mis-labelled. The Bragg peak should be surrounded by CDW satellites, as the upper and lower Bragg peaks are. The pattern shown by arrows in these figures would indicate a first and second CDW harmonic to just one side of the Bragg peak. If the authors did not mis-label, then taking into account this specific second harmonic but not any others would be an inconsistent way of Fourier filtering the data.

3) Interpretation.

Ignoring my reservations about the experimental procedure, I have serious doubts about the interpretation of the data:

- The pattern of strong bonds shown in fig 6c-e is simply not a period-7 CDW. As drawn, this is a period-1 ($Q=0$, nematic) distortion with topological defects (domain walls or dislocation lines). Between the defect lines, the structure is the same in every unit cell, making the pattern clearly $Q=0$ or nematic. Adding domain walls with period 7 is not the same thing as making a period-7 CDW. The difference is clearly visible in the charge modulation itself:

Period-7 CDW: $\rho = \cos(2\pi x / 7 a') \cdot \text{intra-cell structure}$

nematic pattern: $\rho = \text{intra-cell structure} \cdot \text{step function domain walls}$.

In the density patterns shown in figure 6, no $\cos(2\pi x / 7 a')$ component is visible.

- The relation of the observed dimerization to the speculated nesting vector is unclear. If the nesting vector is $\sim 2\pi/7a'$, it should give rise to a $\cos(2\pi x / 7 a')$ density distribution, not dimerization.

- The remark that $b^* \cdot q$ rather than q may induce domain walls and dimerization does not make sense: b^* is a reciprocal lattice vector, and lattice momenta are only defined up to reciprocal lattice vectors. Up to an overall phase shift, a CDW with q and $b^* \cdot q$ thus have the same density modulations.

- The authors model the charge as sitting on the Sb-Sb bonds, but this is not consistent with the data shown in fig 5c-d. The lobes are extended along horizontal and vertical directions, not diagonally as expected for charge delocalized along Sb-Sb bonds. Moreover, the maximum charges never coincide with Sb atoms, and in fact stay close to Te throughout much of the data shown. This further puts into question to what extent the observed pattern can really be attributed to Sb orbitals. Finally,

in fig 5c one of the sides of the unit cell is clearly favoured, whereas fig 5d seems mirror-symmetric. This is not consistent with the pattern of Sb-Sb bond densities drawn in fig 6c.

Because of the above reservations, I cannot recommend publication of this work in Nature Communications.

Reviewer #3

(Remarks to the Author)

The authors study the orbital texture in CeSbTe surface using STM. They revealed that a 1x7 CDW in the normal state just above the antiferromagnetic transition temperature. Through detailed analysis of the anisotropy around the bright spots in the STM images, they uncovered the internal structure of the CDW. They also put forth a model to explain their observations. I find the topic interesting and analysis of STM data seems reasonable. I believe the paper appeal to the audience of Nature Communications, but I would ask the authors to address the following:

In the unit cell-averaged data shown in Fig. 4f and Fig. 5c, the strong intensity appears to be shifted from the high-symmetry points of the crystal, which is confusing. Could this be due to slight anisotropy in the tip? It seems that two domains were measured with the same tip in Fig. S10. I believe performing the same analysis on a 90-degree rotated domain, compared to the one currently shown, would provide a more robust interpretation of the results.

Did the STS or dI/dV mapping measurements reveal the size of the CDW gap or any inversion of the CDW pattern?

Given the doping level estimated by the authors, the crystal should be orthorhombic. Can the anisotropy in the lattice constants be confirmed?

Up to what temperature can this CDW pattern be observed? Establishing a connection between the CDW and the Kondo effect would significantly enhance the quality of the paper.

Version 1:

Reviewer comments:

Reviewer #1

(Remarks to the Author)

I think the authors have fully addressed the issues raised by the reviewers. Therefore, the manuscript in its current form can be published in Nature Communications.

Reviewer #2

(Remarks to the Author)

I thank the authors for their thorough answers to the issues raised by all referees, and for their constructive attitude towards the concerns raised.

The reply of the authors adequately addresses a number of my concerns, and I appreciate the additional information provided by the authors on these points.

Especially the recalibration of the piezo; the addition of tunnelling conductance, R, and Z maps; the discussion of (not observing) contrast inversion and CDW gap; and the added discussion of sample anisotropy, are very much appreciated.

Some concerns still remain, however (detailed below), which prevent me from supporting publication in Nature Communications.

In summary, I believe the data obtained by the authors to be of high quality, and the subject area to be both timely and of inherent interest.

However, I also believe the data is over-interpreted, and the main claim made by the authors of observing a zig-zag CDW pattern in the Sb bonds, is not sufficiently supported by the data. I would support publication of a shorter paper, presenting just the data and its analysis without the speculative interpretation, or perhaps a paper in which it is made clear which conclusions are actually proven by the data, and which are based on assumptions or speculations that are not directly supported by data.

Below, I explain my remaining doubts about the interpretation.

I apologize that this does not follow the numbering in the reply letter.

1) First, for completeness, I would like to respond to the discussion of context:

I appreciate the authors focussing on the similarity of their data with the stripe-checkerboard discussion in the cuprates. However, especially when a new discussion emerges in a high-profile field, it is important to safeguard against making the mistakes that have been made previously in similar situations in less high-profile fields. In this context, the 1980's work of Fukutome on p-orbital order in cubic (= "3D square") lattices of Se and Te (and later work adding Po to the same category) is certainly very relevant to the cuprate discussion. Especially so since the "extra U(1) symmetry missed in the cuprates" is exactly the U(1) relative CDW-phase degree of freedom already introduced by Fukutome and analysed in detail by others since.

I appreciate that the authors recognize that it is also the same physical mechanism suggested to yield chirality in TiSe₂.

2) Then, about the interpretation of the current data:

The proposed microscopic tunneling pathway from Sb-centered states into the STM tip that the authors use to interpret their data at +1V is extremely complex. Fig R1 is argued to show "filtering" of the Sb orbitals through Ce and Te orbitals, but if I understand the caption correctly the figures in panel b really show iso-amplitude surfaces in DFT wavefunctions. I very much appreciate the additional work done by the authors in producing this calculation. However, it does not address the concern raised by myself and reviewer 1: showing one of the wavefunctions that might contribute to the tunneling current, is not the same as calculating the full tunneling current and showing that this one wavefunction provides a dominant contribution to it. Without showing that the combined amplitude of all other possible DFT states between 0 and +1V at the same height as the white positions in panel d is much smaller than the 2.5e-8 arb.units mentioned in the legend, it is not possible to conclude that the authors really observe this specific orbital wavefunction.

In other words: it is not clear from this calculation, nor directly from the data, that the authors really observe the Sb orbitals on which they base their entire interpretation. This problem was raised in different ways by two of the referees, and is not sufficiently resolved in the reply.

3)

If the current work is the first observation of 1x7 CDW order in this compound, that significantly raises the burden of proof for the current authors. Not only do the authors claim to observe a previously unknown CDW phase, they also interpret it in terms of an exotic CDW pattern, based on three-layer deep orbitals that are typically hard to see in STM experiments. Of course this triple claim might be correct. But since the interpretation includes three separate original and exotic ingredients, I would like to insist on direct experimental evidence rather than speculative interpretation.

For the 1x7 pattern: if this has not been previously supported, how do the authors know it is a bulk CDW, rather than a surface reconstruction, or some other spurious effect? The authors do not report contrast inversion, gaps, or other direct evidence of a CDW. I understand these signatures are hard to obtain, and that diffraction or ARPES data is probably not available at the moment. But without any direct evidence, even the starting point that the authors observe a real, bulk CDW at all is not clearly established -- let alone an interpretation that ascribes exotic properties to that CDW.

4)

Even if it agrees with other Fourier filters, I maintain that a Fourier filtering that includes the positions of some second harmonics but not others (as well as some first harmonics and not others) is biased, therefore cannot contain reliable physical information, and should not be shown.

To emphasize: an unbiased Fourier filtering would include intensity in a fixed-radius annulus (or other symmetric shape) around all Bragg reflections, including where no intensity is obviously visible.

5)

The discussion of visible and invisible CDW harmonics in fig R5 might give an explanation for why CDW reflections close to (0,0) are absent, but it does not reproduce the intensities seen in fig 3 of the main text. In particular, the large intensity mismatch of harmonics on either side of the (+-2pi/a,0) reflections is absent. I appreciate that the model presented by the authors is consistent with part of their data. But when suggesting an exotic interpretation of a previously unknown phase of matter, I don't think one can simply ignore disagreement with other equally visible aspects of the same dataset.

6)

The interpretation of nesting vectors in fig R6 cannot be correct. The two vectors (one in panel c, and one in panel e) are in fact the same vector. They connect the same electronic states, just drawn in different Brillouin zones. Since these are the same vectors, connecting the same physical states, they must correspond to the same charge density distortion. The two patterns in panels d and f therefore cannot be distinct. This is a mistaken interpretation of what a nesting vector larger than a reciprocal lattice vector implies for the charge redistribution.

Since the difference between the panels d and f is an essential ingredient of fig 6 in the main text, this further calls into question the interpretation presented by the authors.

Reviewer #3

(Remarks to the Author)

In the revised manuscript and the authors' response to the previous peer review report, all my questions/comments have been addressed. Therefore, I recommend that the manuscript be published in Nature Communications in its current form.

Version 2:

Reviewer comments:

Reviewer #2

(Remarks to the Author)

I thank the authors for their thorough and detailed responses to my second report.

Although I am personally still not fully convinced by the interpretation given by the authors, I recognise that such disagreement among scientists is part of the standard scientific practice, and essential to driving research forward.

Since the authors took care in the resubmitted manuscript to clearly separate their data (which I agree is of high quality and beyond dispute) from their interpretation (including thorough discussion of how the authors arrive at their conclusions), I believe the manuscript now offers readers the opportunity to make an unbiased judgement for themselves. This is all one can expect from a scientific article, and I thus support publication in the present form.

Response to the Reviewers and Revisions

Reviewer #1:

In this manuscript, X. Que *et al.* reported an STM study on the CDW state of $\text{Ce}(\text{Sb}_{1-x}\text{Te}_x)\text{Te}_{1-\delta}$. They observed the anisotropic lobes of the charge density wave in the STM topography at positive bias dominated by Sb p states, while these anisotropic lobes were not observed at negative bias dominated by Te states. Then, in their p_x/p_y bond model in the Sb layers, they attributed these anisotropic lobes to zigzag chains of Sb-Sb bonds. Thus, the authors demonstrate the p -orbital texture in the 1×7 CDW of $\text{Ce}(\text{Sb}_{1-x}\text{Te}_x)\text{Te}_{1-\delta}$. I think the result is interesting, but some ambiguities need to be clarified. I can potentially suggest publication when the issues below are addressed:

Our response:

We thank the reviewer for their positive interest in our work.

Comment 1:

1. The lattice constant of the Te termination measured in this study is 4.14 Å, but the Te-Te distances should be 4.35 Å. Though, due to the thermal drift, STM is not as precise as XRD in measuring the lattice constant, 5% is still a quite big error. I would suggest the authors check the piezo calibration. Or, if the 4.14 Å were true, the optical measurement would be required to confirm this.

Our response:

We thank the reviewer for raising this point. We have checked our piezo calibration using topographic images acquired around the same time period as Fig. 2a of the main text, but on different compounds, 1T-TaS₂ and Ta₂NiSe₅, whose lattice constants are known. Based on an average of eight images, we find that our piezo calibration needs to be rescaled by a factor of 1.06 ± 0.02 . The correct lattice constant of the Te termination should be 4.4 Å. We apologize for the ambiguity caused by our oversight.

Actions taken:

The scale bars of affected topographies have been corrected. The lattice constant of the Te termination is reported as 4.4 Å.

Comment 2:

2. Fig. 3h shows the electrons tunnel to the deep Ce d orbitals and, then, deeper Sb p orbitals at positive bias. And in rest of the work, the authors only focus on discussing Sb p orbitals. Usually, it is different [difficult] to tunnel to the local d orbitals, but, in this case, Sb p orbitals are deeper than Ce d orbitals. I think the authors cannot ignore the contribution of Ce d orbitals in this work.

Our response:

We agree with the reviewer that the role of the Ce d orbitals should be better explained in our manuscript. We believe that the Ce d orbitals play at least two crucial roles:

1. As tunneling filter: Despite the increased contribution of Sb p orbitals to the tunneling current at +1.0 V, the tunneling path originating from Sb p orbitals is still filtered through Ce and remnant Te states (I). Imprints of Ce and Te contributions are evident in the histograms of Figs. 4f and 5c, which show that the anisotropic lobes are not fully centered on the Sb-Sb bonds, but have some extension towards the Ce and Te sites.

More specifically, the filtering of antibonding Sb p_x/p_y orbitals through Ce dx^2-y^2 orbitals, which have lobes directed along the crystallographic a/b directions, may serve to reshape the zigzag lobes of charge density in the Sb layer into more rounded anisotropic lobes elongated along a/b at the surface. We test this idea qualitatively

with DFT slab simulations of a 1×1 CDW state, as shown in Fig. R1. In the 1×1 CDW state, the Sb-Sb bonds show $\sim 8\%$ disproportionation according to the zigzag patterns drawn in Fig. 1f. The simulations cannot reproduce every feature of the more complex 1×7 CDW state, but demonstrates how zigzag Sb p_x/p_y orbitals extend via Ce and remnant Te states to the surface and appear in a simulated STM topography as an elongation of the bright lobes along the a direction (dashed blue ellipse).

Fig. R1: Tunneling paths. **a**, DFT-computed isosurface of partial charge density for a Te-terminated slab with 1×1 CDW, integrated from 0 ($=E_F$) to -0.5 eV. **b**, Partial charge density integrated from 0 to $+1.0$ eV. Several isosurfaces are shown to illustrate the tunneling path from subsurface Sb states. **c** and **d**, Simulations of STM topographies based on the isosurfaces of partial charge density in **a** and **b**. The crystal structure is partially overlaid.

2. As participant in CDW: We have emphasized a simple interpretation of the CDW internal structure as a superposition of Sb p_x/p_y bond density waves. The actual pattern of the bond density waves, however, including their periodic phase slips and relative phases, is highly nontrivial. The actual pattern may be material specific and is likely determined by additional interactions, such as interorbital Sb p_x -Sb p_y and Sb p_x/p_y -Ce dx^2-y^2 hybridization. The role of the rare-earth ion in fixing the Sb CDW pattern has also been suggested in NdSbTe (2).

Actions taken:

We have revised our discussion in the main text (“Microscopic picture”), including Fig. 6a, to highlight the role of Ce d orbitals. Figure R1 has been included in Supplementary note 8.

Comment 3:

3. Considering the authors propose the inter CDW structure found at 1V is p orbital texture, it would be better to clarify the orbital texture in the dI/dV mapping versus energy rather than the topography.

Our response:

We consistently observed the same texture of alternating x/y -anisotropic lobes in various other channels:

1. **dI/dV map** (Fig. R2): Compared to bias-dependent topography $z(x, y, eV)$, which integrates states within an interval $[0, eV]$, the tunneling conductance $g(x, y, eV) = dI/dV(x, y, eV)$ probes states only near eV . We observed x/y -anisotropic lobes starting around +100 mV.

Fig. R2: Tunneling conductance map. **a–g**, Selected energy slices of a 3D spectroscopic map, $g(x, y, eV)$. Setup conditions: -300 mV, 45 pA. Lock-in excitation: 28 mV. **h**, Map of lobe anisotropy extracted from **g**, the $+300$ mV layer. A 2° rotation has been applied to align the lobe axes with the x and y axes. The shaded curve marks a domain boundary.

2. **“ R map”** (Fig. R3): The tunneling current I is proportional not only to the density of states N integrated within $[0, eV]$, but also a setpoint prefactor $f(x, y, z)$ that involves the tunneling matrix elements, apparent barrier height, and exponential decay of the wave function:

$$I(x, y, z, eV) = f(x, y, z) \int_0^{eV} N(x, y, E) dE.$$

To eliminate setpoint effects, Kohsaka *et al.* (3) introduced a ratio map

$$R(x, y, eV) = \frac{I(x, y, z_0, +eV)}{I(x, y, z_0, -eV)} = \frac{\int_0^{eV} N(x, y, E) dE}{\int_{-eV}^0 N(x, y, E) dE}$$

which involves acquiring tunneling currents $I(+eV)$ and $I(-eV)$ at the same tip-sample distance z_0 , such that the setpoint prefactor is removed by division. We observed x/y -anisotropic lobes using $R(x, y, +300 \text{ mV})$.

3. “**Z map**” (Fig. R3): Similarly, the tunneling conductance $g = dI/dV$ also involves the setpoint prefactor:

$$g(x, y, z, eV) = f(x, y, z)N(x, y, eV),$$

which can be removed by computing the ratio (3)

$$Z(x, y, eV) = \frac{g(x, y, z_0, +eV)}{g(x, y, z_0, -eV)} = \frac{N(x, y, eV)}{N(x, y, -eV)}.$$

We observed x/y -anisotropic lobes with $Z(x, y, +300 \text{ mV})$.

Fig. R3: Ratio maps. **a**, R map, defined as the ratio $I(+300 \text{ mV})/I(-300 \text{ mV})$. **b**, Z map, defined as the ratio $g(+300 \text{ mV})/g(-300 \text{ mV})$. **c** and **d**, Maps of lobe anisotropy extracted from **a** and **b**, respectively.

We utilized Multi Pass $-0.5/+1.0/-0.5 \text{ V}$ topographic imaging for our main analysis simply because it was the most practical approach to acquire a large data set (1216×1216 pixels) spanning a large bias range (1.5 V), while minimizing drift.

Actions taken:

Figures R2 and R3 and related discussion have been included in Supplementary note 7.

Comment 4:

(cont.) There may be bias-dependent effects or other effects shown in the data. For example, the red-blue ratio in fig.4e(1V), S8i(1V), S9c(0.5V) and S10b(0.5V) may be different as visualized by my eyes. The red/blue represents the different anisotropic lobes.

Our response:

There should be small differences between the topographies acquired at +0.5 and +1.0 V because the relative proportion of Sb, Ce, and Te states contributing to the tunneling current is bias dependent. Concerning the red-blue ratio, we have computed and compared the number of red and blue anisotropic lobes in each of the images mentioned:

Fig. 4e (+1.0 V)

- Number of lobes parallel to CDW direction: 852 (46%)
- Number of lobes perpendicular to CDW direction: 982 (54%)

Fig. S8i (+1.0 V) (**now Fig. S10**)

- Number of lobes parallel to CDW direction: 175 (47%)
- Number of lobes perpendicular to CDW direction: 196 (53%)

Fig. S9c (+0.5 V): (**now Fig. S11c**) only include majority single-domain region

- Number of lobes parallel to CDW direction: 1699 (44%)
- Number of lobes perpendicular to CDW direction: 2149 (56%)

Inset of Fig. S10b (+0.5 V) (**now Fig. S12b**)

- Number of lobes parallel to CDW direction: 538 (32%)
- Number of lobes perpendicular to CDW direction: 1137 (68%)

We compute a noticeably different red-blue ratio only in the last image, the inset of Fig. S12b. Here, the low spatial resolution of this large-scale topography (~3 pixels per lattice constant) may have skewed our computation of x/y anisotropy based on second derivatives, even with interpolation based on zero padding of the FT.

Actions taken:

We put a note of caveat in the caption of Fig. S12b regarding the anisotropy analysis for this topography, due to its low spatial resolution.

Reviewer #2:

This article describes the observation of a modulated pattern of charge density in (slightly off-stoichiometric) CeSbTe. The authors interpret their findings in terms of two bond density waves in distinct orbital sectors, with a modulated pattern of relative phase shifts.

The experimental findings are timely, relevant to the field, and look interesting. However, the following issues prevent me from recommending this work for publication in Nature Communications:

Our response:

We thank the reviewer for their thoughtful comments and critique, which deserve our full attention and consideration.

Comment 1:

1) Context.

Although perhaps not crucial to the results themselves, I would like to point out that the state of the art is misrepresented in the introduction.

The authors claim that orbital-structured density waves have not received much attention in the literature yet, but they seem to have missed a whole section of the literature that is directly relevant to the results presented here.

As a pointer for where to start looking, I would like to mention at least:

Fukutome, Prog. Theor. Phys. 71, 1 (1984) -- this introduces orbital-structured density waves in p-orbital materials, precisely along the lines used in the present work. There is more work along these lines by the same author, and the orbital-structured density wave in these same p-orbital materials attracted attention from others as well.

Whangbo & Canadell, J. Am. Chem. Soc. 114, 9587 (1992) -- lays the theoretical foundation for the general treatment of multiple-Q density waves in distinct orbital sectors. Again, there is more work by the same authors.

Peng et al., Phys. Rev. Research 4, 033053 (2022) -- a recent example of experimental identification of the same type of orbital-structured charge density wave as the one discussed by the authors in the present work.

Again, these are just three examples of what is an active field of research, whose relevance to the current work ought to be explored by the authors.

Our response:

We have no intention of minimizing or neglecting the body of literature on multi- q and orbitally textured CDW in other materials, such as elemental chalcogenides and transition-metal chalcogenides, particularly the interesting case of $1T$ -TiSe₂, as mentioned by the reviewer. Our statement in the introduction is that orbitally textured CDW in p -electron square lattices has not received as much attention in literature.

The p -electron square lattice has long been known in condensed matter physics as a model system for the infamous “stripes vs. checkerboard” problem: should the ground state be a double- q “checkerboard” CDW that only breaks translational symmetry, or a single- q “stripe” CDW that breaks both translational and rotational symmetries (i.e., smectic)? Kivelson and others (4) have elucidated the conditions for stripe vs. checkerboard CDW, which are also thought to be relevant to the cuprates. Only recently has the realization emerged that the CDW in a p -electron square lattice could break even more symmetries beyond checkerboard or stripes. Experiments have detected additional mirror-symmetry breaking, resulting in an axial CDW amplitude mode (5). A recent preprint (6) pointed out that what was missed was an extra $U(1)$ symmetry associated with the p_x/p_y orbital degree of freedom, in the limit of $t_\pi = t_d = 0$. Put in our language, the superposition of p_x and p_y density waves with different relative phases may result in nontrivial orbital texture that breaks additional symmetries.

Our goal was to provide experimental evidence of microscopic orbital texture in the p -electron square lattice and elucidate its origin as a superposition of p_x and p_y bond-centered CDW with relative phase shift. (We note a contrast to the site-centered CDW proposed by Ref. (5).) Having done so, we agree that the underlying mechanism is reminiscent of those in other materials, including the triple- q CDW in $1T$ -TiSe₂. By varying the relative

amplitudes and phase by which the three q modes should be summed in real space, nontrivial orbital texture with chiral symmetry may be realized in $1T$ -TiSe₂ (7).

Actions taken:

We have expanded our discussion under “Outlook” to mention the connection of our STM-derived microscopic picture of orbital texture in the p -electron square lattice to multi- q and orbitally textured CDW in other materials, particularly $1T$ -TiSe₂, with pertinent references.

Comment 2:

1) Context (cont.)

Secondly, I wonder if the 1×7 density wave reported by the authors has been seen before? The authors mention 1×5 , 1×3 , and 3×3 , but not 1×7 .

If not, then why did the authors study this particular density wave and not one of the other known ones?

Our response:

The 1×7 CDW has not been previously observed in CeSbTe. What caught our attention was the internal pattern of this CDW with alternating x/y anisotropy, which reminded us of the famous charge order pattern in the cuprates, yet could be more readily explained with just p_x and p_y orbitals. We hope to extend our studies in the future to examine and explain how this pattern evolves with the CDW wave vector at different doping levels.

Comment 3:

2) Experimental procedure.

The authors take great care to describe their experimental procedures and data analysis in detail, which is much appreciated.

However, I do have some major worries about the procedure used:

- The authors used a constant-current mode to scan the surface and map the charge density modulations. However, as recently pointed out in arxiv:2406.03294, constant-current mode is not generally suitable for imaging energy-dependent charge modulation patterns in charge density wave materials.

Our response:

We have checked that our observation of alternating x/y -anisotropic lobes is robust in other imaging channels, including dI/dV maps and R/Z ratio maps, which attempt to remove the setpoint prefactor. Please see our response to Comment 3 from Reviewer 1, including Figs. R2 and R3.

Scarfato *et al.* [arxiv:2406.03294, now published (8)] state that constant-current imaging mode is not suitable for identifying (1) the energy of CDW contrast inversion and (2) the CDW gap edge, both of which were not investigated in our work. We agree that more sophisticated analyses are required to address those questions.

Actions taken:

Figures R2 and R3 and related discussion have been included in Supplementary note 7.

Comment 4:

2) Experimental procedure (cont.)

- On top of that, the authors discuss in detail the charge modulations obtained at +1V when tunneling into the third atomic layer, but they also show that already at $-0.5V$ there is a period-7 modulation associated with the top atomic layer. As the tunneling current depends exponentially on distance, it is not clear to me how the authors can be certain that the period-7 pattern they observe at +1V can really be attributed to the Sb atoms in the third layer.

Even if the Te DOS at +1V is very low, it is much closer to the tip. In constant-current mode, it is then not inconceivable that the spatial modulation in even a very small Te DOS can cause the tip-sample distance to vary and thereby impose a period-7 modulation on whatever other features come from the underlying atomic layers.

- According to fig 3g, the DOS of Ce atoms is as large as that of Sb atoms at +1V, but the Ce atoms are in the second atomic layer and the Sb atoms in the third. Why should the STM images at 1V not be dominated by Ce?

Our response:

Please see our response to Comment 2 of Reviewer 1, where we have clarified the tunneling process at +1.0 V. We do not mean to say that the Te and Ce atoms suddenly become transparent and that we directly tunnel from the STM tip to subsurface Sb p_x/p_y orbitals. We mean to say that because of the suppressed partial DOS of Te above E_F , the tunneling current contains increased contributions from deeper layers, including Ce and Sb states. As illustrated in Fig. R1, the elongation of the anisotropic lobes along a/b is inherited from the zigzag lobes of Sb p_x/p_y orbitals filtered through Ce $d_{x^2-y^2}$ orbitals with lobes extended along a/b .

The +1.0 V topography should still have contributions from remnant Te states. We nevertheless do not believe that the Te DOS imposes an artificial $7a$ periodicity onto the +1.0 V topography, because the 1×7 patterns seen at -0.5 and $+1.0$ V are distinct. The pattern at -0.5 V is a weak modulation in amplitude that produces primary peaks $q_{CDW} = 2\pi/(7a)$ in the FT modulus (Fig. 2), which we attribute to the small distortion of Te atoms in the CeTe buffer layer [~ 0.01 Å, according to Refs. (2, 9, 10)]. The pattern at $+1.0$ V is a modulation in the orientation of anisotropic lobes that appears in the FT modulus as only satellite peaks at $q_{Bragg} \pm q_{CDW}$, which we associate with the large bond disproportionation of the Sb square lattice into a complex pattern of zigzag chains [~ 0.4 Å, according to Refs. (2, 9, 10)]. The primary q_{CDW} peaks are notably absent in the $+1.0$ V image. (Please see our response to Comment 6 for further discussion related to FT analysis.)

Actions taken:

We have revised our discussion in the main text (“Microscopic picture”), including Fig. 6a, to highlight the role of Ce d orbitals. Figure R1 has been included in Supplementary note 8.

Comment 5:

2) Experimental procedure (cont.)

- Looking at the FT views in figure 3 as well as the Supplemental Material, the distance between Bragg peaks in the vertical direction seems larger than that in the horizontal direction. This would suggest that the crystal is strained, and has reduced rotational symmetry. It is then not clear whether the modulations observed by the authors are intrinsic to the material, or caused by the imposed strain.

- Finally, both in figure 3 and throughout the Supplemental Material, the peaks to the left and right seem to be mislabelled. The Bragg peak should be surrounded by CDW satellites, as the upper and lower Bragg peaks are. The pattern shown by arrows in these figures would indicate a first and second CDW harmonic to just one side of the Bragg peak. If the authors did not mis-label, then taking into account this specific second harmonic but not any others would be an inconsistent way of Fourier filtering the data.

Our response:

X-ray refinement of powder $\text{CeSb}_{0.63}\text{Te}_{1.37}$ has revealed an orthorhombic distortion of $\sim 1\%$ (10). There is nevertheless general consensus that the driver of CDW in RSbTe is related to Fermi-surface nesting, as evidenced by a monotonic evolution of the CDW wave vector with doping (9–11), rather than orthorhombic strain.

The Bragg peaks along the x direction in the FT modulus of the $+1.0$ V topography are not mislabeled. At this bias voltage, the intensity of the CDW satellites $q_{\text{satellite}} = (\pm 6/7 \times 2\pi/a, 0)$ happens to be much stronger than the intensity at the Bragg peaks $q_{\text{Bragg}} = (\pm 2\pi/a, 0)$.

We applied Fourier filtering around peaks of significant intensity in the FT modulus, rather than according to a preconceived notion of what the CDW structure should be. Experimentally, the dominant satellite peaks around $q_{\text{Bragg}} = (0, \pm 2\pi/a)$ happen to be the first harmonics $(\pm 1/7 \times 2\pi/a, \pm 2\pi/a)$, whereas the dominant satellite peaks around $q_{\text{Bragg}} = (\pm 2\pi/a, 0)$ happen to be the first harmonics $(\pm 6/7 \times 2\pi/a, 0)$ and second harmonics $(\pm 5/7 \times 2\pi/a, 0)$.

We demonstrate nevertheless in Fig. R4 that a different mask with symmetric filtering around the first harmonics ($\pm 6/7 \times 2\pi/a$, 0) and ($\pm 8/7 \times 2\pi/a$, 0) does not change our conclusions.

Fig. R4: Alternative Fourier filtering. **a** and **b**, +1.0 V topography with Fourier filtering around the Bragg peaks (pink circles) and CDW satellite peaks (first harmonics only; black triangles). **c**, Distribution of the positions of the *x*- and *y*-anisotropic lobes within a reference CeSbTe unit cell. **d**, Filtered topography of **a** with red-blue binning applied to demarcate regions of predominant *x* (red) or *y* (blue) anisotropy. We note that **c** and **d** are qualitatively similar to those of Fig. 5 of the main text with a different Fourier filter.

Actions taken:

We have included Fig. R4, the results of Fourier filtering only around the first harmonics, in Supplementary note 6.

Comment 6:

3) Interpretation.

Ignoring my reservations about the experimental procedure, I have serious doubts about the interpretation of the data:

- The pattern of strong bonds shown in fig 6c-e is simply not a period-7 CDW. As drawn, this is a period-1 ($Q=0$, nematic) distortion with topological defects (domain walls or dislocation lines). Between the defect lines, the structure is the same in every unit cell, making the pattern clearly $Q=0$ or nematic. Adding domain walls with period 7 is not the same thing as making a period-7 CDW. The difference is clearly visible in the charge modulation itself:

Period-7 CDW: $\rho = \cos(2\pi \times / 7 a') * \text{intra-cell structure}$

nematic pattern: $\rho = \text{intra-cell structure} * \text{step function domain walls}$.

In the density patterns shown in figure 6, no $\cos(2\pi \times / 7 a')$ component is visible.

Our response:

We thank the reviewer for their comments here. We try to clarify a few key points:

1. The microscopic Sb p_x/p_y model of Fig. 6 seeks to explain the 1×7 CDW pattern seen at +1.0 V, not at -0.5 V.

2. In the -0.5 V topography, the 1×7 CDW signature originates from surface Te atoms and appears as a simple amplitude modulation of $7a$ periodicity, resulting in dominant FT components at $q_{\text{CDW}} = 2\pi/(7a)$ (e.g., Fig. 2)

3. In the $+1.0$ V topography, the 1×7 CDW signature originates from deeper Sb orbitals (with filtering from Ce and remnant Te states) and appears more intricately as alternating x/y -anisotropic lobes of $7a$ periodicity. The dominant FT components lie at $q_{\text{Bragg}} \pm q_{\text{CDW}}$, rather than q_{CDW} .

4. The reduction of the FT component at q_{CDW} in the $+1.0$ V topography comes from the fact that it is the x/y orientation of the anisotropic lobes that modulates with $7a$ periodicity, rather than the amplitude of the lobes. To illustrate, we simulate a superposition of x - and y -anisotropic lobes each modulating with period $7a$, but 180° phase-shifted from each other (Fig. R5). In Fourier space, the q_{CDW} components are summed out of phase and thereby reduced (though not completely zero). The satellite components nevertheless survive, and hence the dominant FT intensity appears at $q_{\text{Bragg}} \pm q_{\text{CDW}}$.

Fig. R5: Simulation of alternating x - and y -anisotropic lobes. **a**, Simulated square lattice of x -anisotropic Gaussian lobes whose amplitudes modulate with $7a$ periodicity. The corresponding FT is shown below. **b**, Simulated square lattice of y -anisotropic Gaussian lobes whose amplitudes modulate with $7a$ periodicity, but phase-shifted by π relative to **a**. The corresponding FT is shown below. **c**, Sum of images **a** and **b**, which resembles the experimental $+1.0$ V topography. The q_{CDW} component in the FT is reduced and the intensity appears first at $q_{\text{Bragg}} \pm q_{\text{CDW}}$.

5. Our microscopic Sb p_x/p_y model is similarly composed of zigzag chains that alternate along the x and y directions with period $7a$. (In the reviewer's terminology, it may be approximately viewed by a local nematic $q = 0$ CDW with periodic phase slips and orthogonal domain boundaries.) This pattern reproduces the FT spectrum of the $+1.0$ V topography with dominant FT components at $q_{\text{Bragg}} \pm q_{\text{CDW}}$, rather than q_{CDW} . To simplify our schematic in Fig. 6, we have only differentiated "strong" and "weak" bonds, i.e., dimers, but in reality, there may be additional small modulation of the dimer strengths, e.g., with component q_{CDW} .

Actions taken:

We have included Fig. R5 and the explanation of why q_{CDW} is reduced in the +1.0 V topography in Supplementary note 5.

Comment 7:

3) Interpretation (cont.)

- The relation of the observed dimerization to the speculated nesting vector is unclear. If the nesting vector is $\sim 2\pi/7a'$, it should give rise to a $\cos(2\pi \times / 7 a')$ density distribution, not dimerization.

- The remark that $b^* \cdot q$ rather than q may induce domain walls and dimerization does not make sense: b^* is a reciprocal lattice vector, and lattice momenta are only defined up to reciprocal lattice vectors. Up to an overall phase shift, a CDW with q and $b^* \cdot q$ thus have the same density modulations.

Our response:

We try to explain the dimer-like pattern of 1×7 bond density waves in another way:

When the p -electron square lattice with nonsymmorphic symmetry (i.e., two atoms per unit cell) is half filled, the Fermi surface comprises a diamond-shaped pocket with nearly parallel segments (Fig. R6a, next page). As explained in the main text, the nesting wave vector of $q = (2\pi/a, 0)$ results in dimerized chains along x' or y' that are phase-shifted relative to each other (Fig. R6b).

Upon electron doping, the Fermi surface splits into two concentric diamond-shaped pockets. Instead of considering the interpocket nesting wave vector $q = (2\pi/(7a), 0)$, let us consider the two intrapocket wave vectors. We view the system instead as two sets of pockets, the smaller one with $q = (2\pi/a - \delta_1, 0)$ and the larger one with $q = (2\pi/a + \delta_2, 0)$ (Figs. R6c–R6f). The combination or interference of these two density waves should result in oscillations with a “fast frequency” of $2\pi/a$, which are the local dimers, and a “slow frequency” of $(\delta_1 + \delta_2)/2 = 2\pi/(7a)$, which is the overall periodicity of the pattern due to phase slips and additional modulation.

Actions taken:

We have included Fig. R6 in Supplementary note 9 and expanded upon our explanation of the bond density wave pattern in the main text (“Microscopic picture”).

Comment 8:

3) Interpretation (cont.)

- The authors model the charge as sitting on the Sb-Sb bonds, but this is not consistent with the data shown in fig 5c-d. The lobes are extended along horizontal and vertical directions, not diagonally as expected for charge delocalized along Sb-Sb bonds. Moreover, the maximum charges never coincide with Sb atoms, and in fact stay close to Te throughout much of the data shown. This further puts into question to what extent the observed pattern can really be attributed to Sb orbitals. Finally, in fig 5c one of the sides of the unit cell is clearly favoured, whereas fig 5d seems mirror-symmetric. This is not consistent with the pattern of Sb-Sb bond densities drawn in fig 6c.

Because of the above reservations, I cannot recommend publication of this work in Nature Communications.

Our response:

We thank the reviewer for raising these points, which deserve a better clarification from us.

We do not equate the anisotropic lobes seen at positive biases with pure Sb p_x/p_y orbitals. They are in fact a convolution of Sb, Ce, and remnant Te states. We are only stating that the anisotropy of these lobes along a/b is inherited microscopically from Sb-Sb zigzag bonds. Therefore, the anisotropic lobes have a complicated spatial distribution within the unit cell that is stretched along the a/b directions, and somewhat shifted towards Sb-Sb bonds, but still with significant weight on Te and Ce atoms.

In Fig. 6, we have only attempted to extract the dominant Sb sector of the 1×7 CDW state from the anisotropic lobes. For this, we map each local minimum in the Fourier-filtered +1.0 V topography to the nearest Sb-Sb bonds, from which we derive the microscopic pattern of Sb p_x and p_y bond density waves. We agree that the anisotropic lobes show additional complexities, which, together with the roles of Ce and Te in the 1×7 CDW state, are still open questions that we seek to better understand.

Actions taken:

We have revised our discussion in the main text (“Microscopic picture”) with the points raised by the reviewer.

Fig. R6. Schematic explanation of p -orbital bond density wave. **a**, Fermi surface of half-filled p -electron square lattice with nesting wave vector $\mathbf{q} = (2\pi/a, 0)$. **b**, Real-space picture of corresponding bond density wave. For simplicity, only the p_y -orbital sector is shown. **c**, Intrapocket wave vector $\mathbf{q} = (2\pi/a - \delta_1, 0)$ of the inner Fermi pocket. **d**, The corresponding bond density wave comprises dimerized chains with periodicity slightly larger than $2a'$, resulting in “missing” dimers (solid circle) when the bonds are properly fixed onto the lattice. **e**, Intrapocket wave vector $\mathbf{q} = (2\pi/a + \delta_2, 0)$ of the outer Fermi pocket. **f**, The corresponding bond density wave comprises dimerized chains with periodicity slightly smaller than $2a'$, resulting in “additional” dimers, i.e., trimers (dashed circle), when the bonds are properly fixed onto the lattice. The 1×7 CDW should be a hybridization of the two patterns in **d** and **f** with both kinds of phase slips.

Reviewer #3:

The authors study the orbital texture in CeSbTe surface using STM. They revealed that a 1×7 CDW in the normal state just above the antiferromagnetic transition temperature. Through detailed analysis of the anisotropy around the bright spots in the STM images, they uncovered the internal structure of the CDW. They also put forth a model to explain their observations. I find the topic interesting and analysis of STM data seems reasonable. I believe the paper appeal to the audience of Nature Communications, but I would ask the authors to address the following:

Our response:

We thank the reviewer for their positive interest in our work.

Comment 1:

In the unit cell-averaged data shown in Fig. 4f and Fig. 5c, the strong intensity appears to be shifted from the high-symmetry points of the crystal, which is confusing. Could this be due to slight anisotropy in the tip? It seems that two domains were measured with the same tip in Fig. S10. I believe performing the same analysis on a 90-degree rotated domain, compared to the one currently shown, would provide a more robust interpretation of the results.

Our response:

The anisotropic lobes are a convolution of Sb, Ce, and remnant Te states, with the anisotropy originating from the zigzag Sb p_x/p_y orbitals, filtered by the Ce dx^2-y^2 orbitals. As such, their local maxima do not perfectly align with Te, Ce, or Sb-Sb bond positions, and therefore appear “shifted” from the high-symmetry points of the crystal.

We note that Figs. 4 and S10 (originally S8) are +1.0 V topographies taken with different tips (reproduced in Figs. R7a and R7b). There are small differences which could be due to tip quality, image resolution, and drift. Nevertheless, the red x -anisotropic lobes consistently spread toward the bottom half of the CeSbTe unit cell and break the mirror plane intersecting the Te atom along the x axis. The blue y -anisotropic lobes do not show any clear violation of mirror symmetry.

What we have noticed is that Fourier filtering, while reducing the spatial distribution of the x and y lobes, does tend to accentuate some additional anisotropies in their spatial distribution, and this has some dependence on the masked FT used (Figs. R7c and R7d). For example, with one masked FT, the red x -anisotropic lobes appear to shift more toward the left side of the unit cell, but with another masked FT with slightly different choice of CDW satellites, they do not.

We regret that we do not have a +1.0 V topography of a 90°-rotated CDW that is precisely matched with a reference Te lattice. We do have a STM dI/dV map matched with a reference Te lattice at lower positive biases (Fig. R7e). The anisotropic lobes parallel to the CDW direction (blue in this case) again break a mirror plane intersecting Te, but the positions of these lobes are shifted further away from Te than the previous images. This small shift could be due to the lower bias voltage used, which means there is a differently weighted combination of Sb, Ce, and remnant Te states contributing to the tunneling current, or we are approaching some inversion from Sb antibonding to bonding states (see response to next comment).

Actions taken:

We have included Fig. R7 in Supplementary note 7. We have clarified which symmetry breaking is robust, and which may be related to Fourier filtering.

Fig. R7. Histogram analysis. **a**, Histograms of x - and y -anisotropic lobes derived from the main +1.0 V STM topography in this work (Fig. 3b). The dashed line with solid arrow represents the violation of a mirror plane. **b**, Histograms derived from an additional +1.0 V topography (Fig. S10). **c** and **d**, Effects of Fourier filtering of **a** with different masks used. **e**, Histograms derived from a +0.1 V STM dI/dV map. Only the majority single-domain region with 90°-rotated CDW (white box) is used.

Comment 2:

Did the STS or dI/dV mapping measurements reveal the size of the CDW gap or any inversion of the CDW pattern?

Our response:

We have considered such questions, but do not have a final answer.

ARPES measurements of a more electron-doped CeSbTe sample with 1×5 CDW have identified a momentum-dependent CDW gap of ~ 0.3 eV (12). In our STM dI/dV measurements, we observe a kink around -0.2 eV (inset of Fig. 3g), but we caution against such a simplistic interpretation. Since CDW gaps are often momentum-dependent, not necessarily gapping out the entire Fermi surface, and not necessarily symmetric about E_F , kink features in dI/dV do not automatically equate to the CDW gap edges, or vice versa [for examples, see Refs. (13, 14)].

We have not observed any obvious signatures of CDW inversion, which we also caution requires more sophisticated analysis and was not the subject of our work [see Ref. (8), suggested by Reviewer 2, as well as Ref. (14)]. We do not have access to the anisotropic lobes at negative biases, where the surface Te p_z states mask out any subsurface contributions from the Sb square lattice. As already mentioned, dI/dV mapping at lower biases (+100 mV) reveal a possible shift in the position of the anisotropic lobes, but other factors could be involved there.

Comment 3:

Given the doping level estimated by the authors, the crystal should be orthorhombic. Can the anisotropy in the lattice constants be confirmed?

Our response:

X-ray refinement of powder $\text{CeSb}_{0.63}\text{Te}_{1.37}$ has revealed an orthorhombic distortion of $\sim 1\%$ (10). We tried to compare the Bragg peaks across orthogonal domains within the same topography, in order to exclude artificial anisotropy related to piezo calibration and drift (Fig. R8). We could not resolve an expected tiny increase/decrease of the Bragg peaks along the x/y -directions from one domain to the other.

Fig. R8. Comparison of Bragg peaks across orthogonal domains. **a**, Large-view topography reproduced from Fig. 2d of the main text. Setpoint: -0.5 V, 80 pA. The FTs of the boxed regions from two orthogonal domains are shown. **b** and **c**, Comparison of FT line cuts along the x and y directions of the sample (shaded lines in insets of **a**). While the CDW peaks (q_{CDW}) shows clear rotation across the domains, the Bragg peaks (q_{Bragg}) show no shift within our image resolution.

Comment 4:

Up to what temperature can this CDW pattern be observed? Establishing a connection between the CDW and the Kondo effect would significantly enhance the quality of the paper.

Our response:

We observed a similar pattern of CDW at 77 K, albeit with lower image quality (Figs. R9a and R9b). We could not obtain a reasonably cleaved surface at room temperature, where a 1×5 CDW at a higher doping level was still observed (12).

Lv *et al.* (15) have suggested that despite the antiferromagnetic ordering of Ce^{3+} moments at 2.6 K, CeSbTe could be on the verge of forming a Kondo lattice, as suggested by a moderately enhanced Sommerfeld coefficient of $\gamma = 41$ mJ/(mol K^2) and resistivity whose temperature dependence is logarithmic. Li *et al.* (12) explain that the CDW in CeSbTe , which reduces the density of states near E_F , could be the reason why the Kondo lattice fails. We find

these indications of weak or proximate Kondo physics interesting, but since we did not observe any Kondo resonance near E_F in STM dI/dV down to 4.2 K (Fig. R9c), we cannot weigh in on this aspect of CeSbTe.

Fig. R9. Additional topographies at 77 K and dI/dV spectra at 4.2 K. **a** and **b**, Topographic images acquired at 77 K with -0.5 and $+0.5$ V biases, respectively. Setpoint current: 30 pA. **c**, STM dI/dV point spectra acquired at 4.2 K. Setup conditions: -50 mV, 100 pA, 2.8 mV lock-in excitation. The black curve is an average of the individual spectra (gray curves)

Actions taken:

In the introduction (“Main”), we added a note that weak or proximate Kondo physics has also been reported in CeSbTe.

Additional corrections

- In the caption of Fig. 3g, we additionally note the 20 mV lock-in excitation voltage used for the dI/dV spectrum.
- *Note added* (end of main text): Upon submission of our manuscript, we noticed an STM work on $\text{GdSb}_{0.87}\text{Te}_{1.11}$ in its tetragonal, non-CDW phase, which also observed x - and y -anisotropic lobes at large positive biases, but with a disordered distribution (16).

References

1. P. Choubey, A. Kreisel, T. Berlijn, B. M. Andersen, P. J. Hirschfeld, Universality of scanning tunneling microscopy in cuprate superconductors, *Phys. Rev. B* **96**, 174523 (2017).
2. T. Salters, F. Orlandi, T. Berry, J. F. Houry, E. Whittaker, P. Manuel, and L. M. Schoop, Charge density wave-templated spin cycloid in topological semimetal $\text{NdSb}_x\text{Te}_{2-x-\delta}$. *Phys Rev Mater* **7**, 044203 (2023).
3. Y. Kohsaka, C. Taylor, K. Fujita, A. Schmidt, C. Lupien, T. Hanaguri, M. Azuma, M. Takano, H. Eisaki, H. Takagi, S. Uchida, J. C. Davis, An Intrinsic Bond-Centered Electronic Glass with Unidirectional Domains in Underdoped Cuprates. *Science* **315**, 1380 (2007).
4. H. Yao, J. A. Robertson, E. A. Kim, S. A. Kivelson, Theory of stripes in quasi-two-dimensional rare-earth tellurides. *Phys. Rev. B* **74**, 245126 (2006).
5. Y. Wang, I. Petrides, G. McNamara, M. M. Hosen, S. Lei, Y.-C. Wu, J. L. Hart, H. Lv, J. Yan, D. Xiao, J. J. Cha, P. Narang, L. M. Schoop, K. S. Burch, Axial Higgs mode detected by quantum pathway interference in RTe_3 . *Nature* **606**, 896 (2022).
6. S. Alekseev, S. A. A. Ghorashi, R. M. Fernandes, J. Cano, Charge density-waves with non-trivial orbital textures in rare earth tritellurides. arXiv:2404.17635v2.
7. J. van Wezel, Chirality and orbital order in charge density waves. *Europhys. Lett.* **96**, 67011 (2011).
8. A. Scarfato, Á. Pásztor, L. Sun, I. Maggio-Aprile, V. Pasquier, T. P. Singar, A. Ørsted, I. Pushkarna, M. Spera, E. Giannini, Ch. Renner, Feedback loop dependent charge density wave imaging by scanning tunneling spectroscopy. *Phys. Rev. B* **110**, 085109 (2024).
9. S. Lei, V. Duppel, J. M. Lippmann, J. Nuss, B. V. Lotsch, L. M. Schoop, Charge Density Waves and Magnetism in Topological Semimetal Candidates $\text{GdSb}_x\text{Te}_{2-x-\delta}$. *Adv. Quantum Technol.* **2**, 1900045 (2019).
10. R. Singha, T. H. Salters, S. M. L. Teicher, S. Lei, J. F. Houry, N. P. Ong, L. M. Schoop, Evolving Devil's Staircase Magnetization from Tunable Charge Density Waves in Nonsymmorphic Dirac Semimetals. *Adv. Mater.* **33**, 2103476 (2021).
11. E. DiMasi, B. Foran, M. C. Aronson, S. Lee, Stability of charge-density waves under continuous variation of band filling in $\text{LaTe}_{2-x}\text{Sb}_x$ ($0 \leq x \leq 1$). *Phys. Rev. B* **54**, 13587-13596 (1996).
12. P. Li, B.J. Lv, Y. Fang, W. Guo, Z.Z. Wu, Y. Wu, D.W. Shen, Y.F. Nie, L. Petaccia, C. Cao, Z.-A. Xu, Y. Liu, Charge density wave and weak Kondo effect in a Dirac semimetal CeSbTe . *Sci. China Phys. Mech.* **64**, 237412 (2021).
13. M. Spera, A. Scarfato, Á. Pásztor, E. Giannini, D. R. Bowler, Ch. Renner, Insight into the Charge Density Wave Gap from Contrast Inversion in Topographic STM Images. *Phys. Rev. Lett.* **125**, 267603 (2020).
14. Á. Pásztor, A. Scarfato, M. Spera, F. Flicker, C. Barreateau, E. Giannini, J. van Wezel, Ch. Renner, Multiband charge density wave exposed in a transition metal dichalcogenide. *Nat Comm.* **12**, 6037 (2021).
15. B. Lv, J. Chen, L. Qiao, J. Ma, X. Yang, M. Li, M. Wang, Q. Tao, Z.-A. Xu, Magnetic and transport properties of low-carrier-density Kondo semimetal CeSbTe . *J. Phys.: Condens. Matter* **31**, 355601 (2019).
16. B. Venkatesan, S.-Y. Guan, J.-T. Chang, S.-B. Chiu, P.-Y. Yang, C.-C. Su, T.-R. Chang, K. Raju, R. Sankar, S. Fongchaiya, M.-W. Chu, C.-S. Chang, G. Chang, H. Lin, A. Del Maestro, Y.-J. Kao, T.-M. Chuang, Direct Visualization of Disorder Driven Electronic Liquid Crystal Phases in Dirac Nodal Line Semimetal GdSbTe . arXiv:2402.18893v2.

Manuscript NCOMMS-24-50191A

Response to the Reviewers and Revisions

Reviewer #2:

I thank the authors for their thorough answers to the issues raised by all referees, and for their constructive attitude towards the concerns raised. The reply of the authors adequately addresses a number of my concerns, and I appreciate the additional information provided by the authors on these points. Especially the recalibration of the piezo; the addition of tunnelling conductance, R , and Z maps; the discussion of (not observing) contrast inversion and CDW gap; and the added discussion of sample anisotropy, are very much appreciated.

Some concerns still remain, however (detailed below), which prevent me from supporting publication in Nature Communications. In summary, I believe the data obtained by the authors to be of high quality, and the subject area to be both timely and of inherent interest. However, I also believe the data is over-interpreted, and the main claim made by the authors of observing a zig-zag CDW pattern in the Sb bonds, is not sufficiently supported by the data. I would support publication of a shorter paper, presenting just the data and its analysis without the speculative interpretation, or perhaps a paper in which it is made clear which conclusions are actually proven by the data, and which are based on assumptions or speculations that are not directly supported by data.

Our response:

We thank and appreciate the reviewer for taking their time to thoroughly review our manuscript a second time.

Our manuscript comprises a result and an interpretation. The result is that by tuning the bias voltage in STM topographic imaging, we reveal the CDW internal structure to comprise anisotropic lobes of charge density that modulate along the x and y directions. Our interpretation is that this intricate pattern arises microscopically from a superposition of Sb p_x and p_y bond density waves. We agree with the reviewer that results could be better distinguished from interpretation. To this end, we have revised our manuscript in the following ways:

Actions taken:

1. We have modified the manuscript title to place greater emphasis on the result: “Visualizing the internal structure of the charge-density-wave state in CeSbTe.”
2. We have subdivided our manuscript into a “Results” section and a “Discussion” section. Our interpretation of the STM data in terms of a p_x/p_y pattern falls into the “Discussion” section.
3. We have gone through our manuscript line-by-line and used appropriate wording such as “interpret” and “interpretation” when describing our picture of bond density waves. These changes are highlighted in red font.

Below, I explain my remaining doubts about the interpretation. I apologize that this does not follow the numbering in the reply letter.

Our response:

We take every comment from the reviewer respectfully and seriously. We stand firmly by our microscopic interpretation of the STM images, but acknowledge that it is our responsibility to convince the reviewer and readers through experimental data, theoretical calculations, and logic that is physically grounded. We will demonstrate that our interpretation is consistent with previous x-ray and electron diffraction data (see response to Comment 3).

Comment 1:

First, for completeness, I would like to respond to the discussion of context: I appreciate the authors focusing on the similarity of their data with the stipe-checkerboard discussion in the cuprates. However, especially when a new discussion emerges in a high-profile field, it is important to safeguard against making the mistakes that have been made previously in similar situations in less high-profile fields. In this context, the 1980’s work of Fukutome on p -orbital order in cubic (= “3D square”) lattices of Se and Te (and later work adding Po to the same category) is

certainly very relevant to the cuprate discussion. Especially so since the “extra $U(1)$ symmetry missed in the cuprates” is exactly the $U(1)$ relative CDW-phase degree of freedom already introduced by Fukutome and analysed in detail by others since. I appreciate that the authors recognize that it is also the same physical mechanism suggested to yield chirality in TiSe_2 .

Our response:

In the course of these revisions, we have gained a better appreciation of the vast literature on CDW, thanks to the reviewer. We strive to represent the state-of-the-art as accurately as possible and have no intention of discounting “less high-profile fields,” which is often a subjective measure not reflective of their actual importance.

Comment 2:

Then, about the interpretation of the current data: The proposed microscopic tunneling pathway from Sb-centered states into the STM tip that the authors use to interpret their data at +1V is extremely complex. Fig R1 is argued to show “filtering” of the Sb orbitals through Ce and Te orbitals, but if I understand the caption correctly the figures in panel b really show iso-amplitude surfaces in DFT wavefunctions. I very much appreciate the additional work done by the authors in producing this calculation. However, it does not address the concern raised by myself and reviewer 1: showing one of the wavefunctions that might contribute to the tunneling current, is not the same as calculating the full tunneling current and showing that this one wavefunction provides a dominant contribution to it. Without showing that the combined amplitude of all other possible DFT states between 0 and +1V at the same height as the white positions in panel d is much smaller than the $2.5\text{e-}8$ arb. units mentioned in the legend, it is not possible to conclude that the authors really observe this specific orbital wavefunction. In other words: it is not clear from this calculation, nor directly from the data, that the authors really observe the Sb orbitals on which they base their entire interpretation. This problem was raised in different ways by two of the referees, and is not sufficiently resolved in the reply.

Our response:

We think there is a misunderstanding here. What we have computed with DFT in Figs. 3h and 6a (previous version) and R1b and R1d (previous response) is the charge density from all states integrated between 0 ($=E_F$) and +1.0 eV, i.e.,

$$\rho_{\text{int}}(x, y, z, +1.0 \text{ eV}) = \int_0^{+1.0 \text{ eV}} N(x, y, z, E) dE, \quad (\text{R}'1)$$

where $N(x, y, z, E)$ is the local density of states (LDOS) derived from the Kohn-Sham orbitals. We did not cherry-pick a single specific wave function to favor our interpretation. Likewise, the STM simulation in Fig. R1d (previous response) represents the full tunneling current with contributions from all states within 0 and +1.0 eV.

Figure R'1 is a revised presentation of our calculations, which we hope could better clarify our procedures. For a CeSbTe slab with 1×1 CDW, we calculated the integrated LDOS from 0 to +1.0 eV, as defined in Eq. (R'1). As ρ_{int} is a function of three variables (x, y, z), one convenient way to visualize this quantity in 3D is to plot a series of iso-amplitude surfaces, $\rho_{\text{int}}(x, y, z, +1.0 \text{ eV}) = \rho_0$, for different values of ρ_0 . Figure R'1c shows four such isosurfaces in sequence of decreasing $\rho_0 = 2.5 \times 10^{-3}$, 9.5×10^{-4} , 5×10^{-4} , and 2.5×10^{-8} arb. units. The isosurface at $\rho_0 = 2.5 \times 10^{-3}$ arb. units reveals that the integrated LDOS has highest intensity in the Sb plane and comprises zigzag lobes of the antibonding Sb p_x/p_y orbitals. The isosurfaces at $\rho_0 = 9.5 \times 10^{-4}$ and 5×10^{-4} arb. units reveal that as the integrated LDOS decreases and spreads toward the surface, it picks up contributions from Ce dx^2-y^2 and remnant Te p_z orbitals. To simulate a constant-current STM topography, which is proportional to an isosurface of constant integrated LDOS, we need to pick ρ_0 small enough, such that the average height of the isosurface lies within the vacuum region of the slab. Figure R'1d shows this simulation for $\rho_0 = 2.5 \times 10^{-8}$ arb. units, which corresponds to an average height $\sim 4.8 \text{ \AA}$, which is a typical distance of the tip-sample junction. The simulated +1.0 V topography indeed contains contributions from Te and Ce states, but because the surface Te DOS is largely suppressed in this energy range, fingerprints of the antibonding Sb p_x/p_y orbitals filter through and appear as an anisotropic elongation of the bright lobes (encircled by blue ellipses in Figs. R'1c–R'1e) and a redistribution of their weight away from the Te atomic positions. This is consistent with experiments. The anisotropic lobes are visible for other values of ρ_0 , e.g., 1×10^{-8} arb. units (Fig. R'1e).

Fig. R'1: DFT simulations. **a–e**, Results from a CeSbTe slab *with* 1×1 CDW. **a**, LDOS integrated from −0.5 to 0 eV ($=E_F$), i.e., $\rho_{\text{int}}(x, y, z, -0.5 \text{ eV})$. Four isosurfaces of the same integrated LDOS are shown. **b**, Simulation of −0.5 V STM topography based on the isosurface in **a** at 2.5×10^{-8} arb. units. The height is referenced from the surface Te atom of the slab. **c**, Isosurfaces of the LDOS integrated from 0 to +1.0 eV, i.e., $\rho_{\text{int}}(x, y, z, +1.0 \text{ eV})$. Two views of the isosurface at 2.5×10^{-3} arb. units are shown. **d** and **e**, Simulations of +1.0 V STM topographies for two different isosurfaces of the integrated LDOS in **c** at 2.5×10^{-8} and 1×10^{-8} arb. units, respectively. Signatures of the Sb zigzag antibonding orbitals filter through into the simulations (blue ellipses). **f–i**, Results from a CeSbTe slab *without* CDW. **f**, Isosurfaces of the LDOS integrated from −0.5 to 0 eV. **g**, Simulation of −0.5 V STM

topography based on the isosurface in **f** at 2.5×10^{-8} arb. units. **h**, Isosurfaces of the LDOS integrated from 0 to +1.0 eV. **i**, Simulation of +1.0 V STM topography based on the isosurface in **h** at 2.5×10^{-8} arb. units.

To further confirm that the anisotropic elongation of the bright lobes originates from subsurface Sb p_x/p_y antibonding orbitals, and not something else, we performed two control calculations:

(1) Simulation of -0.5 V STM topography of the 1×1 CDW (Figs. R'1a and R'1b): The isosurfaces show that the integrated LDOS from 0 to -0.5 eV contains sizable contributions from surface Te p_z orbitals already at $\rho_0 = 2.5 \times 10^{-3}$ arb. units, which masks any subsurface contributions. The resulting topography therefore reveals only a square lattice of isotropic bright lobes centered at Te atomic positions, with no signatures of the Sb bond dimers. This is consistent with experiments.

(2) Simulation of +1.0 V STM topography *without* CDW (Figs. R'1h and R'1i): The isosurfaces show that although the Sb p_x/p_y orbitals contribute significantly to the integrated LDOS, in the absence of CDW, their lobes are isotropic in shape and do not produce any distinctive features in the topography.

Actions taken:

We have fixed our explanations in the main text, Methods, and Supplementary note 7 to clarify that the integrated LDOS and STM simulations include all states between 0 and +1.0 eV. We have updated Fig. 6a and replaced Fig. S17 (previous version) with the more detailed Fig. R'1 (now as Fig. S15). We would argue that the DFT simulations provide solid theoretical support for our microscopic interpretation.

Comment 3:

If the current work is the first observation of 1×7 CDW order in this compound, that significantly raises the burden of proof for the current authors. Not only do the authors claim to observe a previously unknown CDW phase, they also interpret it in terms of an exotic CDW pattern, based on three-layer deep orbitals that are typically hard to see in STM experiments. Of course this triple claim might be correct. But since the interpretation includes three separate original and exotic ingredients, I would like to insist on direct experimental evidence rather than speculative interpretation.

For the 1×7 pattern: if this has not been previously supported, how do the authors know it is a bulk CDW, rather than a surface reconstruction, or some other spurious effect? The authors do not report contrast inversion, gaps, or other direct evidence of a CDW. I understand these signatures are hard to obtain, and that diffraction or ARPES data is probably not available at the moment. But without any direct evidence, even the starting point that the authors observe a real, bulk CDW at all is not clearly established – let alone an interpretation that ascribes exotic properties to that CDW.

Our response:

The reviewer identifies three ingredients that form the basis of our interpretation. We were not trying to devise an original or exotic explanation based on a triple leap-of-faith; in fact, our approach was to take the most prudent explanation based on the principles of STM and the known material science of $RSbTe$ and pnictide-/chalcogen-based square lattices. We discuss each of these three points below and supply experimental data from literature.

1. STM can probe contributions from subsurface layers: This is not uncommon to STM experiments. We mention a few prominent examples:

a. STM detection of subsurface charge order and superconductivity. Figure R'2 illustrates how the *same* pattern of charge order can be imaged in two distinct cuprate superconductors, $Ca_{1-x}Na_xCuO_2Cl_2$ (Na-CCOC) and $Bi_2Sr_2Dy_xCa_{1-x}Cu_2O_{8+\delta}$ (Dy-Bi2212), even though the CuO_2 layer with charge order lies in the second layer from the surface in Na-CCOC, and in the third layer from the surface in Dy-Bi2212, and the overlayers of the two compounds are chemically distinct. This imaging is possible because the CaCl, BiO, and SrO overlayers are insulating. If the bias voltage of tunneling lies within the insulating gaps of those overlayers, then the next subsurface layer, CuO_2 , contributes observable features to the STM image. This is the same mechanism by which we detect fingerprints of subsurface CDW in CeSbTe.

Fig. R'2: Examples (I), subsurface tunneling in cuprates. **a** and **b**, Distinct crystal structures of Na-CCOC and Dy-Bi2212 with electronically active CuO_2 plane two or three layers deep, respectively. **c** and **d**, Similar STM dI/dV spectra with “V” shape originating from the subsurface CuO_2 layer. **e** and **f**, Similar images of charge order originating from the subsurface CuO_2 layer. **a–f** are reproduced from Ref. (1) (redacted). **g** and **h**, Theoretical simulations demonstrating how the dx^2-y^2 Wannier function of the subsurface CuO_2 layer maintains its character even above the surface, due to the minimal contributions from the insulating overlayers. **i** and **j**, Simulation of STM dI/dV spectra near a nonmagnetic impurity, showing how the line shape is dominated by the subsurface CuO_2 layer with little difference due to the different overlayers. **g–j** are reproduced from Ref. (2). Reprinted figure with permission from P. Choubey, A. Kreisel, T. Berlijn, B. M. Andersen, P. J. Hirschfeld, Universality of scanning tunneling microscopy in cuprate superconductors, *Phys. Rev. B* **96**, 174523 (2017). Copyright 2025 by the American Physical Society.

b. STM detection of subsurface defects. Figure R'3 illustrates STM imaging of the classic “clover” defects in topological insulators. These defects can be as deep as 5–6 atomic layers.

Fig. R'3: Examples (II), STM visualization of subsurface defects in topological insulators. **a**, $\text{Sb}_{\text{Te}1}$ antisite defects (labeled V) in Sb_2Te_3 that are five atomic layers deep perturb the LDOS at the surface via $pp\sigma$ bonds (shaded pink rectangles), resulting in a characteristic “clover” signature in the STM topography. Reproduced from

Ref. (3). Reprinted figure with permission from Y. Jiang, Y. Y. Sun, M. Chen, Y. Wang, Z. Li, C. Song, K. He, L. Wang, X. Chen, Q.-K. Xue, X. Ma, S. B. Zhang, Fermi-Level Tuning of Epitaxial Sb_2Te_3 Thin Films on Graphene by Regulating Intrinsic Defects and Substrate Transfer Doping. *Phys. Rev. Lett.* **108**, 066809 (2012). Copyright 2025 by the American Physical Society. **b**, STM imaging of Fe interstitial defects in Bi_2Se_3 that are six atomic layers deep, which also produce “clover” signatures. Reproduced from Ref. (4). Reprinted figure with permission from M. M. Yee, Z.-H. Zhu, A. Soumyanarayanan, Y. He, C.-L. Song, E. Pomjakushina, Z. Salman, A. Kanigel, K. Segawa, Y. Ando, J. E. Hoffman, Spin-polarized quantum well states on $\text{Bi}_{2-x}\text{Fe}_x\text{Se}_3$. *Phys. Rev. B* **91**, 161306(R) (2015). Copyright 2025 by the American Physical Society.

c. STM imaging of Moiré superlattices. Figure R’4 shows examples where Moiré superlattices are observed by STM, even when the twisted interface is buried beneath the surface.

Fig. R’4: Examples (III), STM visualization of Moiré patterns from twisted subsurface layers. a, STM topography of twisted double bilayer graphene with a 1.05° -rotated interface that is two atomic layers deep. Reproduced from Ref. (5) (redacted). **b**, STM topography of Bi_2Te_3 , where the first quintuple layer (QL) is rotated 1.2° from the second QL, resulting in Moiré patterns. Reproduced from Ref. (6). Adapted from K. Schouteden, Z. Li, T. Chen, F. Song, B. Partoens, C. Van Haesendonck, K. Park, Moiré superlattices at the topological insulator Bi_2Te_3 . *Sci. Rep.* **6**, 20278 (2016) under a CC BY license: <https://creativecommons.org/licenses/by/4.0/>.

2. The CDW comprises a pattern of zigzag Sb-Sb bonds: We would respectfully refute the notion that we are proposing “an exotic interpretation of a previously unknown phase of matter.” Our interpretation follows from the Zintl-Klemm concept and hypervalent bonding in chemistry applied to p -electron square lattices (7, 8). Sb holds a formal oxidation state of -1 in CeSbTe , which means that it is formally isoelectronic to Te, which is known to form dimers, zigzag chains, and ring structures in covalently bonded networks. When the Sb^{1-} square lattice is doped with electrons in $\text{Ce}(\text{Sb}_{1-x}\text{Te}_x)\text{Te}$, the extra electrons occupy the antibonding states of Sb^{1-} , which has a destabilizing effect on the lattice. The system has to compensate by forming dimers, such that stabilizing lone pairs can be localized in the “gaps” between the dimers. (Canadell makes similar points when discussing CDW in the tellurides (9).) Our microscopic interpretation bridges the reciprocal-space picture of nesting, which determines the preferred CDW periodicity, with the real-space picture of dimerization based on hypervalent bonding.

Figure R’5 presents a comparison of the 1×7 CDW state, which was determined by our STM measurements, with the bulk 1×3 and 1×5 CDW states of $\text{Ce}(\text{Sb}_{1-x}\text{Te}_x)\text{Te}_{1-\delta}$, which were experimentally refined from x-ray diffraction (10). All three CDW patterns comprise zigzag chains of Sb-Sb bonds. Furthermore, we illustrate how each of these CDW patterns can be decomposed into a superposition of p_x and p_y bond density waves. The bond density waves are composed of dimerized chains that (1) are shifted from neighboring chains by a' and (2) contain periodic phase slips. In the 1×3 CDW pattern, the periodic phase slip consists of two consecutive strong bonds, i.e., a trimer. In the 1×5 CDW pattern, the periodic phase slip consists of two consecutive weak bonds. It is natural that the 1×7 CDW pattern contains both kinds of phase slips. The unified picture of these three CDW states strengthens the case for our microscopic interpretation.

Fig. R'5: Consistency of CDW internal structure measured by x-ray diffraction and STM. **a** and **b**, Crystal structures of $\text{Ce}(\text{Sb}_{1-x}\text{Te}_x)\text{Te}_{1-\delta}$ samples with 1×3 and 1×5 bulk CDW states, respectively, as refined by x-ray diffraction. Reproduced from Ref. (10) (redacted). **c** and **d**, Decomposition of 1×3 and 1×5 CDW internal structures into superpositions of Sb p_x and p_y bond density waves. **e**, Decomposition of 1×7 CDW internal structure, as determined by STM, into a superposition of Sb p_x and p_y bond density waves. Reproduced from Fig. 6 of main text.

3. The 1×7 pattern is a CDW: Given that RSbTe is famous for hosting a variety of CDW states, it is more prudent to interpret our observations in terms of CDW, rather than to invoke an unknown surface electronic state. We unfortunately do not have diffraction and ARPES data on the same $\text{Ce}(\text{Sb}_{1-x}\text{Te}_x)\text{Te}_{1-\delta}$ crystals that we measured with STM. We note, however, that in $\text{Gd}(\text{Sb}_{1-x}\text{Te}_x)\text{Te}_{1-\delta}$, bulk CDW states that are larger than 1×5 , including 1×6 CDW (11) and 1×7 CDW (unpublished data from co-author S.L.), have been observed (Fig. R'6a). There is no fundamental reason against a CDW larger than 1×5 in $\text{Ce}(\text{Sb}_{1-x}\text{Te}_x)\text{Te}_{1-\delta}$, other than the fact that a high-quality x-ray refinement becomes harder due to the increasing complexity, as well as disorder and domains.

Figures R'6b and R'6c compare our $+1.0$ V STM Fourier transform with electron diffraction data for CDW in a $\text{LaSb}_{0.7}\text{Te}_{1.2}$ crystal (12). We note the striking similarity of these patterns in that the dominant superstructure peaks do not occur at the fundamental wave vectors \mathbf{q}_{CDW} , but at the satellites $\mathbf{q}_{\text{Bragg}} \pm \mathbf{q}_{\text{CDW}}$. The authors identify the

dominant superstructure peak as $q_{\text{Bragg}} - q_{\text{CDW}} = 0.836 \times (2\pi/a)$, which is close to $\sim 2\pi[5/(6a)]$, i.e., a $\sim 1 \times 6$ CDW state. Likewise, we identify a dominant superstructure peak $q_{\text{Bragg}} - q_{\text{CDW}} = 2\pi[6/(7a)]$ in the +1.0 V STM topography of a 1×7 CDW state. The similarity of the reciprocal-space patterns suggests that electron diffraction and STM are probing the same internal structure of CDW, only that STM offers additional access to phase information.

Fig. R'6: Bulk CDW with longer wavelengths in $R(\text{Sb}_{1-x}\text{Te}_x)\text{Te}_{1-\delta}$. **a**, CDW wave vectors q in $\text{Gd}(\text{Sb}_{1-x}\text{Te}_x)\text{Te}_{1-\delta}$ as function of doping x , as determined from x-ray diffraction. Λ refers to integer multiples of the lattice constant. Reproduced from Ref. (11) with unpublished data from co-author S.L. (redacted) **b**, Selected area electron diffraction (SAED) of a $\text{LaSb}_{0.7}\text{Te}_{1.2}$ crystal aligned near the [001] axis. The dominant superstructure wave vector, $q = 0.836a^* \approx (5/6)a^*$, implies a 1×6 CDW. Reproduced from Ref. (12). Reprinted figure with permission from E. DiMasi, B. Foran, M. C. Aronson, S. Lee, Stability of charge-density waves under continuous variation of band filling in $\text{LaTe}_{2-x}\text{Sb}_x$ ($0 \leq x \leq 1$). *Phys. Rev. B* **54**, 13587-13596 (1996). Copyright 2025 by the American Physical Society. **c**, For comparison, the Fourier transform amplitude of our STM +1.0 V topography is shown, showing similar dominant satellite peaks $q = 0.858a^* \approx (6/7)a^*$.

Actions taken:

Figure R'5 and related discussion have been included in Supplementary note 9.

Comment 4:

Even if it agrees with other Fourier filters, I maintain that a Fourier filtering that includes the positions of some second harmonics but not others (as well as some first harmonics and not others) is biased, therefore cannot contain reliable physical information, and should not be shown.

To emphasize: an unbiased Fourier filtering would include intensity in a fixed-radius annulus (or other symmetric shape) around all Bragg reflections, including where no intensity is obviously visible.

Actions taken:

In the main text, we have shown the filtered pattern with twofold symmetric masks around the first-harmonic satellites, as requested by the reviewer. We have updated Figs. 5, 6b, S7, and S8 accordingly.

Comment 5:

The discussion of visible and invisible CDW harmonics in fig R5 might give an explanation for why CDW reflections close to (0,0) are absent, but it does not reproduce the intensities seen in fig 3 of the main text. In particular, the large intensity mismatch of harmonics on either side of the $(-2\pi/a, 0)$ reflections is absent. I appreciate that the model presented by the authors is consistent with part of their data. But when suggesting an

exotic interpretation of a previously unknown phase of matter, I don't think one can simply ignore disagreement with other equally visible aspects of the same dataset.

Our response:

Our only intent with Fig. R5 was to address the specific question of how the FT intensity at q_{CDW} could be suppressed by summing two out-of-phase modulations. The microscopic model we want to propose was presented in Fig. 6, not Fig. R5. We apologize for the confusion.

Regarding the intensity mismatch of the first harmonics, we note that this effect is reproducible across different tips and topographies (e.g., compare Figs. S7 and S8). We think it may arise from tunneling matrix elements of the states involved.

Actions taken:

We have removed this Fig. R5 from Supplementary note 5 to avoid confusion.

Comment 6:

The interpretation of nesting vectors in fig R6 cannot be correct. The two vectors (one in panel c, and one in panel e) are in fact the same vector. They connect the same electronic states, just drawn in different Brillouin zones. Since these are the same vectors, connecting the same physical states, they must correspond to the same charge density distortion. The two patterns in panels d and f therefore cannot be distinct. This is a mistaken interpretation of what a nesting vector larger than a reciprocal lattice vector implies for the charge redistribution. Since the difference between the panels d and f is an essential ingredient of fig 6 in the main text, this further calls into question the interpretation presented by the authors.

Our response:

The wave vectors drawn in Fig. R6 (previous response) correspond to $\mathbf{b}^* - \mathbf{q}_{\text{CDW}} = 2\pi[6/(7a)]$ and $\mathbf{b}^* + \mathbf{q}_{\text{CDW}} = 2\pi[8/(7a)]$. They indeed connect the same states (in different Brillouin zones) and both give rise to 1×7 CDW; however, they represent distinct harmonics of the 1×7 CDW pattern. We simply wanted to highlight the role of higher harmonics in the actual 1×7 CDW internal structure.

We recognize that the reviewer raised a similar issue in the previous round, so we strive again to improve our explanation. The fact that our Fermi surface has well-nested pockets separated by $q = 2\pi/(7a)$ implies that the system can gain electronic energy if the lattice distorts and adopts a new periodic potential $V'(x)$ with $7a$ periodicity, i.e., $V'(x + 7a) = V(x)$. This does not imply, however, that $V'(x)$ should be as simple as $\cos[2\pi/(7a)x]$. The form of this new periodic potential, as well as the CDW internal structure, depend specifically on material details, such as the electron-phonon coupling and local bonding tendencies. In particular, both STM and electron diffraction experiments (Fig. R'6) confirm that certain satellites such as $\mathbf{b}^* - \mathbf{q}_{\text{CDW}}$ have more dominant intensity than the fundamental wave vector \mathbf{q}_{CDW} . We rationalize this experimental fact as follows:

Suppose we began with a 1×1 CDW in stoichiometric CeSbTe. We slightly dope the system $\text{Ce}(\text{Sb}_{1-x}\text{Te}_x)\text{Te}$ such that the Fermi surface splits into two concentric diamonds separated by $q = 2\pi/(20a)$, thereby stabilizing a hypothetical 1×20 CDW. We think it is unphysical for the original bond density wave with $\cos[(2\pi/a)x]$ modulation to abruptly transform into a pure $\cos[2\pi/(20a)x]$ modulation with large reorganization of charge density. Instead, we suggest that the small doping simply weakens the original dimerized bonds and creates periodic phase slips, such that the effective periodicity of the new state is 1×20 . In this pattern, certain harmonics, such as $\cos[2\pi(19/(20a))x]$ or $\cos[2\pi(21/(20a))x]$, which closely resemble the original $\cos[(2\pi/a)x]$ modulation, could be much more dominant than the $\cos[2\pi/(20a)x]$ modulation. It is then natural to draw nesting wave vectors corresponding to $\mathbf{b}^* \pm \mathbf{q}_{\text{CDW}}$, rather than to \mathbf{q}_{CDW} itself. (Canadell similarly describes the 1×3 CDW in EuTe_4 with the wave vector $\mathbf{b}^* - \mathbf{q}_{\text{CDW}} = (2/3)\mathbf{b}^*$ (9).)

The $\mathbf{b}^* \pm \mathbf{q}_{\text{CDW}}$ wave vectors also illustrate how the 1×7 CDW state is continuously connected to the 1×1 CDW state. Upon moderate electron doping, the Fermi surface splits into two concentric diamonds, and apparently the nesting wave vector undergoes a discontinuous change from the *intrapocket* wave vector $q = 2\pi/a$ to the *interpocket* wave vector $q = 2\pi/(7a)$. We are simply suggesting that we could arrive at the same conclusion of a

1×7 CDW state by tracking the two *intrapocket* wave vectors, which evolve continuously to $q = 2\pi(6/7a)$ and $2\pi(8/7a)$.

Actions taken:

The essential ingredient of the microscopic picture in Fig. 6 is the STM data, not our discussion of nesting. In the main text, we rationalize this microscopic pattern based on electron doping and destabilization of dimerized bonds, leading to phase slips in real space. Our discussion of nesting wave vectors is now found solely in an updated Supplementary note 8. There, we emphasize that these wave vectors all result in a 1×7 CDW, but we simply wish to highlight the role of higher harmonics in the internal bond-density-wave pattern.

References

1. Y. Kohsaka, C. Taylor, K. Fujita, A. Schmidt, C. Lupien, T. Hanaguri, M. Azuma, M. Takano, H. Eisaki, H. Takagi, S. Uchida, J. C. Davis, An Intrinsic Bond-Centered Electronic Glass with Unidirectional Domains in Underdoped Cuprates. *Science* **315**, 1380 (2007).
2. P. Choubey, A. Kreisel, T. Berlijn, B. M. Andersen, P. J. Hirschfeld, Universality of scanning tunneling microscopy in cuprate superconductors, *Phys. Rev. B* **96**, 174523 (2017).
3. Y. Jiang, Y. Y. Sun, M. Chen, Y. Wang, Z. Li, C. Song, K. He, L. Wang, X. Chen, Q.-K. Xue, X. Ma, S. B. Zhang, Fermi-Level Tuning of Epitaxial Sb₂Te₃ Thin Films on Graphene by Regulating Intrinsic Defects and Substrate Transfer Doping. *Phys. Rev. Lett.* **108**, 066809 (2012).
4. M. M. Yee, Z.-H. Zhu, A. Soumyanarayanan, Y. He, C.-L. Song, E. Pomjakushina, Z. Salman, A. Kanigel, K. Segawa, Y. Ando, J. E. Hoffman, Spin-polarized quantum well states on Bi_{2-x}Fe_xSe₃. *Phys. Rev. B* **91**, 161306(R) (2015).
5. C. Rubio-Verdú, S. Turkel, Y. Song, L. Klebl, R. Samajdar, M. S. Scheurer, J. W. F. Venderbos, K. Watanabe, T. Taniguchi, H. Ochoa, L. Xian, D. M. Kennes, R. M. Fernandes, Á. Rubio & A. N. Pasupathy, Moiré nematic phase in twisted double bilayer graphene. *Nat. Phys.* **18**, 196 (2022).
6. K. Schouteden, Z. Li, T. Chen, F. Song, B. Partoens, C. Van Haesendonck, K. Park, Moiré superlattices at the topological insulator Bi₂Te₃. *Sci. Rep.* **6**, 20278 (2016).
7. G. A. Papoian and R. Hoffmann, Hypervalent Bonding in One, Two, and Three Dimensions: Extending the Zintl-Klemm Concept to Nonclassical Electron-Rich Networks. *Angew. Chem. Int. Ed.* **39**, 2408 (2000).
8. S. Klemenz, A. K. Hay, S. M. L. Teicher, A. Topp, J. Cano, L. M. Schoop, The Role of Delocalized Chemical Bonding in Square-Net-Based Topological Semimetals. *J. Am. Chem. Soc.* **142**, 6350 (2020).
9. J.-P. Pouget and E. Canadell, Structural approach to charge density waves in low-dimensional systems: electronic instability and chemical bonding. *Rep. Prog. Phys.* **87**, 026501 (2024).
10. R. Singha, T. H. Salters, S. M. L. Teicher, S. Lei, J. F. Khoury, N. P. Ong, L. M. Schoop, Evolving Devil's Staircase Magnetization from Tunable Charge Density Waves in Nonsymmorphic Dirac Semimetals. *Adv. Mater.* **33**, 2103476 (2021).
11. S. Lei, S. M. L. Teicher, A. Topp, K. Cai, J. Lin, G. Cheng, T. H. Salters, F. Rodolakis, J. L. McChesney, S. Lapidus, N. Yao, M. Krivenkov, D. Marchenko, A. Varykhalov, C. R. Ast, R. Car, J. Cano, M. G. Vergniory, N. P. Ong, L. M. Schoop, Band Engineering of Dirac Semimetals Using Charge Density Waves. *Adv. Mater.* **33**, 2101591 (2021).
12. E. DiMasi, B. Foran, M. C. Aronson, S. Lee, Stability of charge-density waves under continuous variation of band filling in LaTe_{2-x}Sb_x (0 ≤ x ≤ 1). *Phys. Rev. B* **54**, 13587-13596 (1996).